# Ultra-deep sequencing reveals dramatic alteration of organellar genomes in *Physcomitrella patens* due to biased asymmetric recombination

Masaki Odahara [1,4,5✉], Kensuke Nakamura[2,5], Yasuhiko Sekine[1] & Taku Oshima [3✉]

Destabilization of organelle genomes causes organelle dysfunction that appears as abnormal growth in plants and diseases in human. In plants, loss of the bacterial-type homologous recombination repair (HRR) factors RECA and RECG induces organelle genome instability. In this study, we show the landscape of organelle genome instability in *Physcomitrella patens* HRR knockout mutants by deep sequencing in combination with informatics approaches. Genome-wide maps of rearrangement positions in the organelle genomes, which exhibited prominent mutant-specific patterns, were highly biased in terms of direction and location and often associated with dramatic variation in read depth. The rearrangements were location-dependent and mostly derived from the asymmetric products of microhomology-mediated recombination. Our results provide an overall picture of organelle-specific gross genomic rearrangements in the HRR mutants, and suggest that chloroplasts and mitochondria share common mechanisms for replication-related rearrangements.

[1] Department of Life Science, Rikkyo (St. Paul's) University, Tokyo, Japan. [2] Department of Life Science and Informatics, Maebashi Institute of Technology, Gunma, Japan. [3] Department of Biotechnology, Toyama Prefectural University, Toyama, Japan. [4] Present address: Biomacromolecules Research Team, RIKEN Center for Sustainable Resource Science, Saitama, Japan. [5] These authors contributed equally: Masaki Odahara, Kensuke Nakamura. ✉email: masaki.odahara@riken.jp; taku@pu-toyama.ac.jp

Chloroplasts (cp) and mitochondria (mt) are organelles of endosymbiotic origin that conduct photosynthesis and respiration, respectively. To perform their energy production functions, both of these organelles have their own genomic DNA and gene expression system. The organelle DNAs are smaller than the genomes of the ancestral bacteria due to gene transfer to the nucleus over the course of evolution. In general, land plant cpDNA is mapped as a ~100–200 kb circular structure, with large single-copy (LSC) and small single-copy (SSC) regions separated by a pair of large inverted repeat (IR) regions. Land plant mtDNA is also mapped as a circular structure and varies in size from 50 kb to 11.7 Mb[1]. They are, however, actually shown to form linear, branched, and circular structures in many plants[2]. The organelle genomes are functionally essential because they encode genes for photosynthesis, respiration, and organelle gene expression; moreover, because they are present in the cell in multiple copies, they can also serve as reservoirs of nutrients[3]. Although more than 30 years have passed since the first sequences of plant cpDNA[4] and mtDNA[5] became available, the dynamics of these genomes, including their replication, remain incompletely understood. In several species, cpDNA has been proposed to replicate via a D-loop as well as via a rolling-circle model[6]. Rolling-circle replication has also been proposed for plant mtDNA, but the existence of complicated multimeric forms of both mtDNA[7] and cpDNA[8] suggest that the organelle genomes replicate in a manner that depends on homologous recombination. On the other hand, recombination also involves dynamics of organelle genomes, especially mitochondrial genomes. Recombination between long repeats (>1 kb), which are often observed in angiosperm mitochondrial genomes, yields reciprocal products that sometimes associate with production of subgenomes, making mtDNA into a multichromosomal structure[9]. Recombination between intermediate-size repeats (<500 bp) leads to accumulation of asymmetric recombination products, a hallmark of plant mitochondrial genome dynamics[10,11]. Frequent recombination between the large IRs produces two types of isomeric cpDNA structures in which the SSC is inverted[12].

Recent studies have reported several genes and proteins that influence the stability of organelle genomes in the moss *Physcomitrella patens*[13] and the angiosperm *Arabidopsis thaliana*[14]. *P. patens* is a basal land plant that shows exceptionally high efficiency of nuclear gene targeting by homologous recombination[15]. Its genomes have been sequenced: (nuclear DNA, 511 Mb[16,17]; cpDNA, 123 kb[18]; and mtDNA, 105 kb)[19]. *P. patens* nuclear DNA encodes two homologs of the bacterial-type RecA recombinase involved in homologous recombination repair (HRR); the gene products of *RECA1* and *RECA2* are localized to mitochondria and chloroplasts, respectively[20,21]. Knockout of *RECA1* leads to severe growth defects and gross rearrangement of mtDNA caused by aberrant recombination between short dispersed repeats (SDRs; <100 bp) and dosage imbalance of mtDNA[22]. Knockout of *RECA2* results in a modest growth defect, chloroplast genome instability due to induced recombination between SDRs, and a moderate decrease in the overall level of cpDNA[23]. *A. thaliana* has three RECAs, which are localized to the chloroplast (AtRECA1), chloroplast and mitochondria (AtRECA2), and mitochondria (AtRECA3)[10]. Note that gene names and localization of their products as well as their phylogenetic relationship do not correspond between *P. patens* and *A. thaliana* regarding *RECA* genes[13,24]. At*RECA1* mutants exhibit a slight increase in cpDNA rearrangements with or without microhomology[25]. At*RECA2* or At*RECA3* is involved in the maintenance of mitochondrial genome stability by suppressing aberrant recombination between SDRs, while At*RECA3* has only minor involvement in the suppression[26].

RECG, a homolog of RecG DNA helicase, which is involved in HRR in bacteria, is a dual-targeted protein encoded by *P. patens* nuclear DNA. *RECG* KO mutants exhibit a growth defect similar to but milder than that caused by *RECA1* KO, and they also exhibit mitochondrial genome instability due to more frequent recombination between SDRs, which is sometimes associated with genome rearrangements as in *RECA1* KO mutants[27]. On the other hand, the SDRs that involve mitochondrial genome instability in *RECG* KO mutants differ from those in *RECA1* KO mutants, and the gene dosage imbalance of mtDNA caused by *RECG* KO also exhibits a different pattern from that caused by *RECA1* KO. Furthermore, *RECG* KO mutants exhibit chloroplast genome instability due to induction of recombination between SDRs, which are more prominent than those of *RECA2* KO mutants[23,27,28]. *RECG* KO mutants have higher copy numbers of every cpDNA locus, whereas *RECA2* KO mutants have slightly lower copy numbers of all cpDNA loci. The *A. thaliana* RecG homolog, RECG1, is localized to both chloroplasts and mitochondria, as with *P. patens* RECG. *A. thaliana RECG1* mutants accumulate elevated levels of products from recombination between mitochondrial SDRs; the chloroplast genome stability of the *RECG1* mutants has not yet been characterized[29]. Collectively, the discovery of organelle HRR factors in these divergent species suggest a ubiquitous mechanism for the maintenance of organelle genome stability in plants.

Interestingly, *P. patens RECA1* and *RECG* mutants exhibit asymmetric accumulation of recombination products, which defines biased accumulation of one of the two types of reciprocal recombination products, in mitochondria[22,27], as in *A. thaliana* mutants of *MutS homolog 1* (*MSH1*)[30], which also participates in maintenance of organelle genome stability by suppressing aberrant recombination between SDRs in both *A. thaliana*[30,31] and *P. patens*[28]. This phenomenon is frequently observed as the consequence of recombination between intermediate-size repeats in angiosperm mitochondria[32]. However, the nature of such biased recombination at the genome level remains unclear in both mitochondria and chloroplasts.

Recently, it became possible to detect genome rearrangements using next-generation sequencing (NGS) data. Two approaches have been applied to this task: paired-end mapping and junction point detection in chimeric reads. For example, the Pindel software uses paired-end information to detect indels within a user-specified size[33], and a method called PRISM (pair-read informed split mapping) uses the combination of pair-read and split-read information to detect various structural variants[34]. In order to understand the role of HRR in the maintenance of organelle genomes, we performed deep sequencing analysis of cpDNA and mtDNA on the Illumina NGS platform to comprehensively and thoroughly identify the genomic rearrangements. Using the Illumina NGS data, we followed two independent approaches to detect the rearrangements, junction read (split read) mapping and paired-read mapping on the genome. The deep sequencing data revealed the distribution in read depth of organelle DNA, which are implicated as the consequences of genome instability and genome rearrangement. Our results show that the HRR system strongly represses the locus-dependent and extremely biased recombination of organelle genomes.

## Results

**Dynamic alteration of organelle DNA abundance in HRR mutant strains.** Using total *P. patens* genomic DNA of wild type (WT), *RECA1* KO, *RECA2* KO, and *RECG* KO lines, we prepared libraries for Illumina sequencing. To avoid biased amplification at AT-rich regions, which are frequently observed in *P. patens* cpDNA, the libraries were prepared without PCR in order. Illumina sequencing of genomic DNA (150 bp paired-ended reads) yielded 9–11 Gb of reads for each strain (Supplementary

Table 1). In our computational analysis of split reads, first, we mapped the reads using immap, which uses simple index to search for the most appropriate mapping position of reads allowing up to 75 mismatches per read while allowing no gapped alignment[35]. In WT, mapping to the reference sequences of the nuclear[17], chloroplast[18], and mitochondrial[19] genomes yielded 12.5-fold, 27,000-fold, and 3900-fold average coverage, respectively (Table 1). The composition of the mapped reads was 63.1% ncDNA, 32.5% cpDNA, and 4.0% mitochondrial DNA (mtDNA) (Fig. 1 and Supplementary Table 2). Neither strict mapping allowing two mismatches per read, nor mapping of the truncated paired-end reads (first 100 bp) changed the proportion of reads mapped to each genome (Supplementary Fig. 1a and b). Considering the effect of sequencing error, a maximum of 2 bp mismatches were allowed in the strict mapping, which improved mapping ratio substantially while ensuring mapping accuracy (Supplementary Data 1). The number of reads mapped for cpDNA revealed little or moderate increase in *RECA1* and *RECG* KO mutants as compared to the WT, whereas an apparent decrease was observed in *RECA2* KO mutants (Fig. 1, Supplementary Fig. 1, and Supplementary Table 2). By contrast, the mapping rate of mtDNA was slightly elevated in *RECA1* KO lines and largely elevated in *RECG* KO lines. The dynamic changes in the abundance of cpDNA and mtDNA are consistent with previous analyses of organelle DNA loci in these mutants[23,27].

**Table 1 Sequencing data for genome analysis of WT *Physcomitrella patens*.**

|  | [a]Nucleus | [b]Chloroplast | [c]Mitochondrion |
|---|---|---|---|
| Genome size (bp) | 511,000,000 | 122,890 | 105,340 |
| Mapped reads (bp) | 6,398,772,300 | 3,296,285,850 | 409,282,950 |
| Coverage | 12.5 | 26823.1 | 3885.4 |

[a]Schween, Gorr[16], [b]Sugiura, Kobayashi[18], [c]Terasawa, Odahara[19].

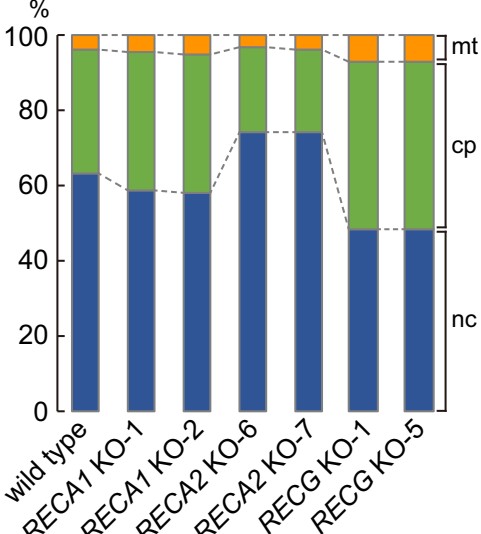

**Fig. 1 Ratio of mapped reads.** Ratios of reads mapped to nuclear (nc), chloroplast (cp), or mitochondrial (mt) DNA by allowing 75 maximum mismatches in a 150-bp read (h75r10, the mapping parameters of NGS analysis explained in Fig. 4) are shown as percentages of total reads. The source data for the graphs are found in Supplementary Data 4.

**Instability of organelle genomic DNA in *RECA* and *RECG* mutants.** cpDNA contains identical IR sequence regions. Because the reads assigned to these regions cannot be distinguished, one of the repeats (pos. 113,302–122,890) was removed from the reference cpDNA for the purposes of mapping; accordingly, the mapping depth of the remaining IR region (pos. 85,212–94,800) is doubled (Supplementary Fig. 2). In the WT, the depth of cpDNA remained constant, except in the IR region (Supplementary Fig. 2). The copy numbers through the entire organelle genome indicated alterations of the DNA dosages in specific mtDNA loci in *RECA1* KO lines, and in both mtDNA and cpDNA in *RECG* KO lines, whereas the organelle DNA dosage of *RECA2* KO lines were indistinguishable from those of the WT (Supplementary Fig. 2 and Supplementary Table 3). To plot the net amount of organelle DNA, we divided the depth of mutants by the depth of WT at each window, and then normalized with the number of reads mapped on mutant and WT, for each organelle. This procedure resulted in a smooth and sensitive display of organelle DNA abundance (Fig. 2). The copy numbers of *RECA2* KO cpDNA, in LSC and SSC, were moderately reduced (Fig. 2 C1 and C2), as was predicted based on our previous findings[23]. The copy number of cpDNA in the *RECA2* KO mutant was not constant throughout the chloroplast genome; the variation in copy number relative to the WT genome formed gentle slopes, with peaks at pos. 40,000 and 100,000, and a minimum at pos. 80,000 (Fig. 2 C1 and C2), suggesting that cpDNA was abnormally replicated in the *RECA2* KO mutant. Knockout of *RECG* leads to greater abundance of cpDNA loci in both LSC and SSC[27], resulting in a substantial variation in cpDNA dosage: doses in IR, SSC, and LSC (pos. 0–50,000) were elevated, whereas dosage in pos. 42,000–85,000 remained comparable to the WT (Fig. 2 C3 and C4).

The copy number of WT mtDNA remained constant throughout the entire genome, as did that of cpDNA (Supplementary Fig. 2). Consistent with our previous quantification of three mtDNA loci[27], the depth of mtDNA in *RECA1* mutant (Fig. 2 M1 and M2) was higher in the first half of the genome, and lower in the second half, than that of the wild type, especially around pos. 60,000, explaining the severe morphological defects in *RECA1* KO plants[22]. In addition, knockout of *RECA1* also led to abrupt increases in copy number at pos. 16,000 and 85,000 in one of the two lines (*RECA1* KO-2 in Fig. 2 M2). Knockout of *RECG*, which increases the copy number of the three mtDNA loci[27], resulted in an increase in copy number in the first half of mtDNA relative to WT, and the dosage gradually decreased from pos. 1 to pos. 100,000 (Fig. 2 M3 and M4). These copy number variations in the organelle genome were observed in two independent mutant lines and confirmed by quantitative PCR (Supplementary Fig. 3), ruling out the possibility of an experimental artifact. These data suggest that the lack of HRR induces genome instability, resulting in dramatic alteration of organelle DNA structure.

**Computational detection of rearrangements from Illumina reads.** To detect genomic rearrangements from the Illumina NGS data, we followed two independent approaches: paired-read mapping on the genome (Fig. 3a), and junction (split) read (Fig. 3b) methods (Fig. 4 and Supplementary Fig. 4). We call reads including a rearrangement within the 150 bp stretch as Junction Reads. The blue part of the junction read matches with the reference at position A and the orange part matches at A′, as shown in Fig. 3b. To locate these two mapping positions for each junction read, we utilized the mapping method we developed, that does not allow gapped alignment, and allows mismatches as many as half the base pairs per read, while it guarantees the assigned

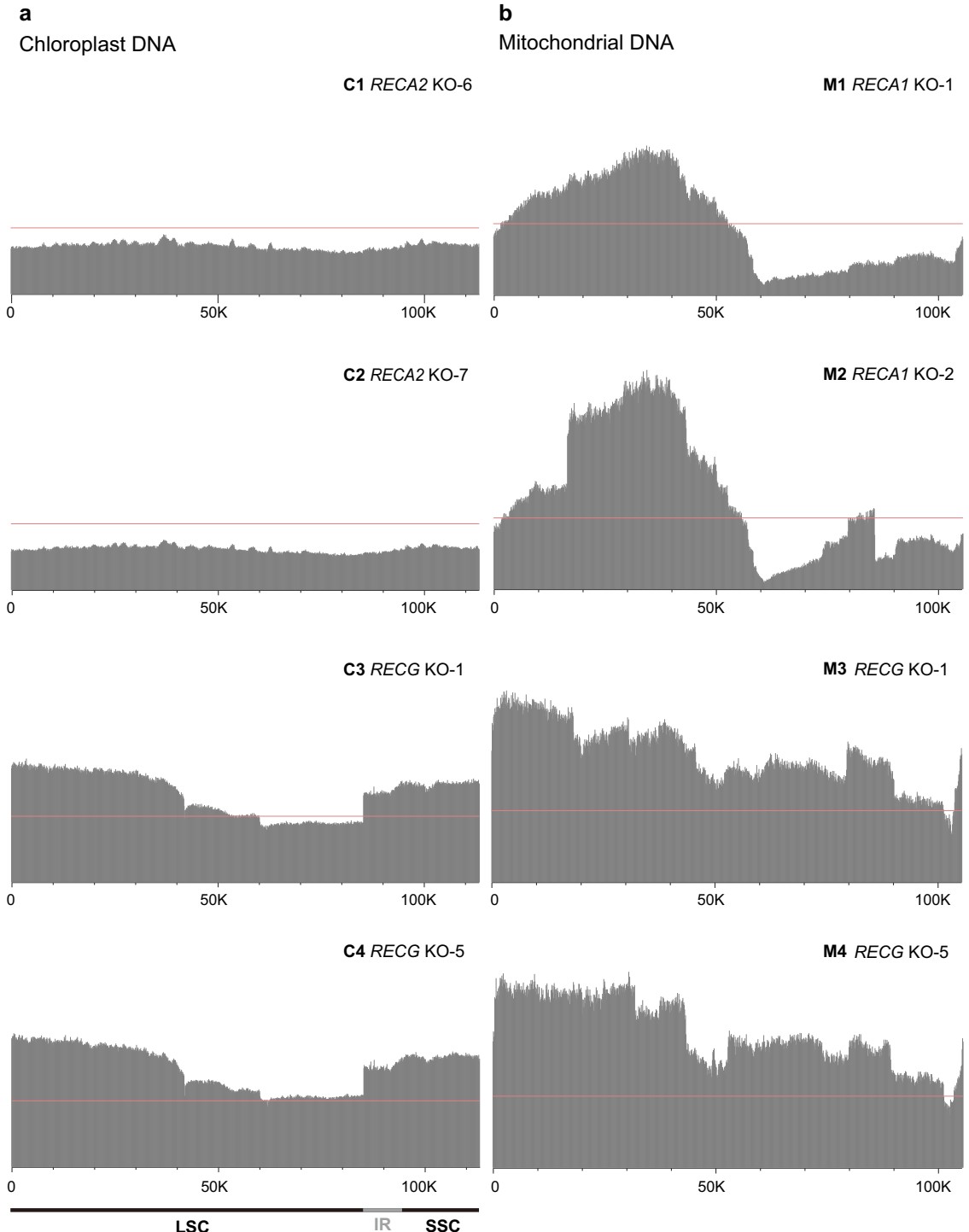

**a** Chloroplast DNA

**b** Mitochondrial DNA

**Fig. 2 Mapping depth of mutant organelle DNA.** Mapping results of mutant chloroplast DNA (**a**) and mitochondrial DNA (**b**). Mapped reads of mutant cpDNA and mtDNA were normalized against those of WT cpDNA and mtDNA, respectively. Red line indicates the average depth expected from relative copy number among cpDNA and mtDNA in WT. One copy of the large inverted repeats was removed from the cpDNA map.

position is the position with least mismatches for the read on the reference. With this method, junction reads are mapped on the position where the longer end of the segment matches (Fig. 3b blue part). Artificial mapping results such as those of contaminated sequences, can be distinguished from junction reads, as mismatches for such misalignments distribute evenly along the read sequence. After the mapping, reads with mismatch-free region on one side (blue), and mismatches-prone region on the other end (orange), are collected as junction read candidate.

Junction read candidates sharing the same junction point on the reference are gathered to form a junction read cluster and the consensus sequence of the mismatch prone region is taken and the position where this consensus sequence matches is located as the rearrange position (Fig. 3b′ and Supplementary Fig. 5). To re-evaluate each junction read cluster, we excluded reads with a mismatch in the longer (blue) segment, or more than one mismatch in the shorter (orange) segment. We also excluded junction reads when their paired read is not mapped on the same reference

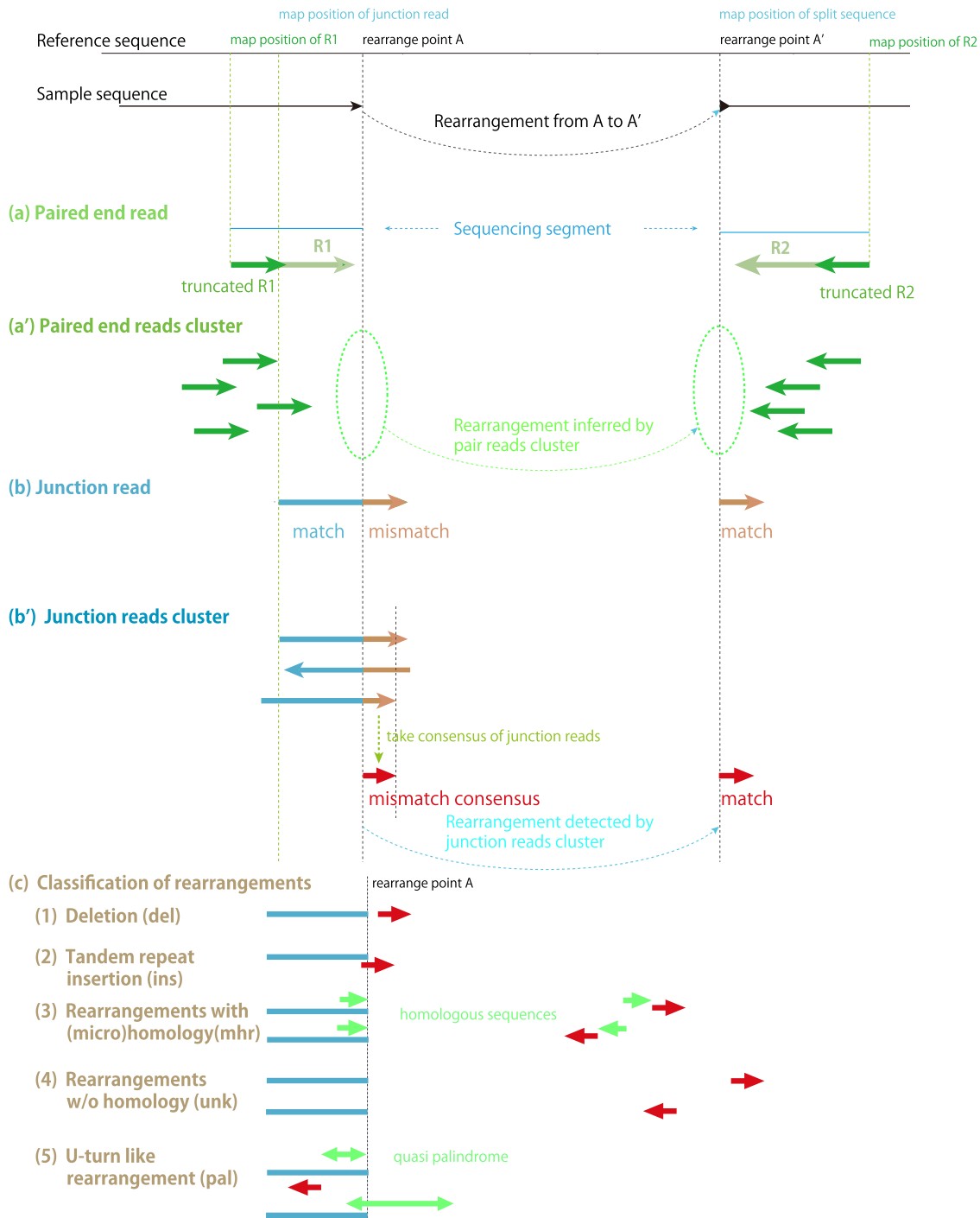

**Fig. 3 Detection of rearrangement by (a) paired-end reads and (b) junction reads.** A rearrangement within a sequencing segment can be detected when the distance between the mapped positions of paired-end reads, R1 and R2, is longer than expected or pointing toward the same direction (**a**). Paired reads sharing locations in close vicinity are collected to determine the presence and position of rearrangement (**a**'). The rearrangement can be detected as a junction read when one side of the read matches one position in the genome, and the other side matches a different position (**b**). Junction reads sharing the same boundary are collected as a junction cluster, and the consensus of mismatch sequence is searched on genome to locate the rearrange position (**b**'). Based on the relative positions and sequences around them, rearrangements are classified into five categories (**c**). (1) Deletion (del), (2) Tandem repeat insertion (ins), (3) Rearrangements caused by microhomology (mhr), (4) Rearrangements without sequence similarity (unk), and (5) U-turn like rearrangement (pal).

sequence, or not mapped on appropriate position relative to the positions of the latter end (arrowhead) of the junction read.

Junction clusters are then categorized into five groups regarding the type of rearrangement (Fig. 3c). They include short INDELs because we do not allow gapped alignment while mapping.

INDELs shorter than 50 bps are categorized as (1) Deletion (del) and (2) Insertion (ins). Note here we only detect tandem repeat for insertions, but insertions of arbitrary sequence. Rearrangement between two distant locations on reference are categorized as (3) Rearrangement based on microhomology/homology (mhr) if there

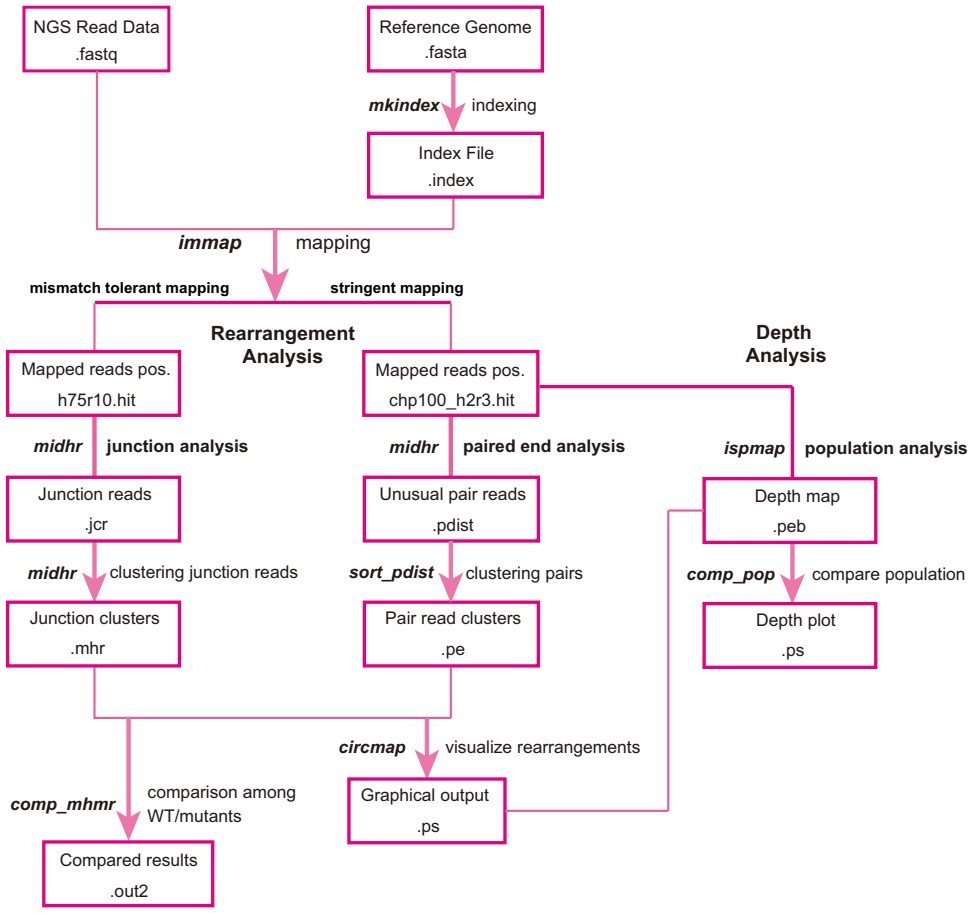

**Fig. 4 Flow chart of the computational analysis.** Reads are mapped to the reference genomes by immap in two ways. For the analysis of rearrangements using paired-end reads, truncated (100 bp) reads mapped under stringent conditions are analyzed by midhr to identify rearrangements with unusual positions. Identified unusual pairs are subsequently clustered by sort_pdist and visualized by circmap. For the analysis of rearrangements using junction reads, reads (150 bp) mapped under mismatch-tolerant conditions are analyzed by midhr to identify and create junction clusters, and the clustered junction reads are visualized by circmap. The results of paired-end reads and junction reads are compared between strains by comp_mhmr.

are any identical sequences longer than 3 bps responsible for the rearrangement, and (4) Rearrangement without any sequence similarity (unk), if there is no homologous sequence around the junction points. Depending on the direction of homologous sequences, mhr can be further categorized as same or opposite direction (Supplementary Fig. 6). Because a pair of repeat can cause two sets of rearrangements (Supplementary Fig. 6), we unified mhr data associated with the same pair of repeats as mhmr data, thereby the number of two type of rearrangements associated with one pair of repeats can be compared (Supplementary Fig. 6). Detailed procedure to form mhmr dataset is described in the Methods. Rearrangements within close vicinity (<70 bps) in opposite direction are categorized as (5) U-turn like rearrangements (pal; Supplementary Fig. 7). The sequence for these rearrangements can be observed as a palindrome, or quasi palindrome sequence. Supplementary Data 3 lists all rearrangements (ins, del, mhr, unk, pal, mhmr) detected in our analysis.

Number of the junction clusters and the junction reads are summarized in Table 2. Matching ratio of the longer and the shorter segments of junction reads and pair reads of the junction reads are summarized in Supplementary Table 4. The overall matching ratio of junction reads is more than 99.9% and matching ratio of pair reads of junction reads is more than 96%. Some of the paired reads may include rearrangements same/different as the corresponding junction read, and that is the reason why the mapping ratio of the paired reads is slightly lower.

Nevertheless, the high matching ratio suggests rearrangements detected by our junction read method are reliable.

Paired-end mapping, which used information from paired-end reads, has been used previously to detect the structural variants of the genome when paired reads are arranged in an unusual manner[36], and we utilized this method to support the junction reads analysis. In the paired end analysis, we truncated each pair reads from 150 bp to 100 bp (Fig. 3a). We carried out mapping for the paired end analysis carefully by permitting 2 bp mismatch in the 100 bp read, so that reads cannot be mapped if they include the rearrangement within the sequence. Truncation of reads allows the reads including junction point in their tails to be mapped, making it possible to detect the rearrangement occurring in the pale green region of Fig. 3a, i.e., 100 bp wider detection range compared to the mapping of original untruncated reads. After mapping these truncated reads, we picked up read pairs separated farther than expected from the average insert length, and read pairs pointing toward the same direction. These paired reads imply the presence of rearrangement between them. Then read pairs sharing mapped positions in close vicinity are put together to form a paired end read cluster (Fig. 3a'). Each paired end read cluster suggests a rearrangement. In the example, the direction of sequence does not change during the rearrangement, therefore the directions of paired reads are facing to each other. On the other hand, if the direction of sequence alter during the rearrangement, as is in the case of U-turn rearrangements

**Table 2 Number of classified junction clusters and junction reads (in parentheses).**

| | aCategory | WT | RECA1 KO-1 | RECA1 KO-2 | RECA2 KO-6 | RECA2 KO-7 | RECG KO-1 | RECG KO-5 |
|---|---|---|---|---|---|---|---|---|
| | | **Strains** | | | | | | |
| Chloroplast | mhr | 11 (62) | 9 (74) | 10 (48) | 40 (329) | 34 (356) | 172 (8196) | 142 (9148) |
| | pal | 4 (16) | 10 (47) | 3 (15) | 3 (12) | 2 (8) | 28 (134) | 15 (74) |
| | bunk | 1 (5) | 0 (0) | 1 (3) | 1 (3) | 1 (6) | 5 (381) | 5 (1187) |
| | ins | 31 (351) | 36 (303) | 38 (348) | 31 (209) | 40 (289) | 32 (505) | 36 (307) |
| | del | 258 (3458) | 240 (3331) | 274 (4050) | 180 (1988) | 184 (1859) | 316 (5151) | 302 (5823) |
| | total | 305 (3892) | 295 (3755) | 326 (4464) | 255 (2541) | 261 (2518) | 553 (14,367) | 500 (16,539) |
| Mitochondrion | mhr | 0 (0) | 242 (3097) | 278 (6230) | 0 (0) | 0 (0) | 375 (9401) | 229 (10,361) |
| | pal | 0 (0) | 0 (0) | 0 (0) | 0 (0) | 0 (0) | 1 (4) | 2 (10) |
| | unk | 0 (0) | 0 (0) | 3 (13) | 0 (0) | 0 (0) | 7 (74) | 2 (16) |
| | ins | 5 (29) | 29 (222) | 25 (300) | 5 (23) | 8 (51) | 20 (106) | 22 (218) |
| | del | 15 (90) | 28 (222) | 33 (253) | 5 (28) | 10 (42) | 49 (313) | 43 (578) |
| | total | 20 (119) | 299 (3541) | 339 (6796) | 10 (51) | 18 (93) | 452 (9898) | 298 (11,183) |

The sums of junction reads that formed the junction clusters are shown.
Values are shown as integers.
aCategory of junction reads: mhr, rearrangement with (micro)homology; pal, U-turn like rearrangement; unk, rearrangement without sequence similarity; ins, insertion; del, deletion.
bJunction reads related to the chloroplast large inverted repeats were removed from the read numbers.

described later (Figs. 3c–5), the paired reads will point toward the same direction on the reference.

By the paired end analysis, we can only identify the presence and approximate position of rearrangement (Fig. 3a′) and cannot identify short indels. Also, a paired end read cluster may include more than two independent rearrangements in close vicinity. Nevertheless, paired end analysis was carried out in order to support the results of junction cluster method. We show number of reads in junction read clusters and in paired read clusters, for some rearrangements as examples, in Supplementary Table 5.

**Rearranged DNA products by junction reads and paired-end reads.** As a result of the two kinds of computational analyses, we obtained an overview of the organelle genome rearrangements caused by mutation of *RECA1*, *RECA2*, or *RECG*. For junction read analysis, a 150 bp read can identify the recombination sites in most cases, because *P. patens* organelle DNA contains no repeats longer than 79 bp except for the large IR in cpDNA[22,23] (Supplementary Table 6 and Supplementary Data 2). We identified 410,000–1,010,000 and 51,000–129,000 junction reads for cpDNA and mtDNA, respectively, from each mutant strain (Supplementary Table 7). As a consequence, for each strain, 260–510 and 4–330 clusters were generated to imply specific rearrangement positions for cpDNA and mtDNA, respectively (Supplementary Table 7). The number of unusual pairs derived from rearranged loci was largely elevated in the mutants, especially in cpDNA of *RECG* KO mutants and mtDNA of *RECA1* KO and *RECG* KO mutants (Supplementary Table 8). Arcs in the right columns of Supplementary Figs. 9 and 10 show the links (rearrangements) suggested by the clusters of unusual pairs.

All types of junction read clusters were listed with the number of reads (Supplementary Data 3), and the results were compared with the outcomes of the paired-end read analysis described in the previous section. In terms of positions and frequency, links (rearrangements) predicted by junction reads exhibited patterns similar to those of links inferred by paired-end reads, reinforcing the reliability of junction reads method (Supplementary Fig. 9–11 and Supplementary Table 8).

Next, we compared the data obtained by junction read cluster analyses with data from previous organelle DNA analyses of *RECA1*, *RECA2*, and *RECG* KO mutants carried out by DNA gel blot or quantitative PCR. Knocking out *RECA1*, *RECA2*, or *RECG* leads to accumulation of products from recombination between SDRs, and the involvement of SDRs sometimes differs among these mutants[13]. Moreover, for given recombination products, the accumulation levels often differ between mutants[22,27,28]. As shown in Supplementary Table 9, most previously characterized recombination products were identified as junction clusters, and, in most cases, the presence or absence and accumulation levels of the products analyzed by qPCR or DNA gel blot agreed with the results of the junction read analysis. In particular, the number of junction read clusters corresponding to products of recombination between repeats R9 (*ccmF-atp9*) or R10 (*nad2-atp9*) (Supplementary Data 2), which are characteristic mtDNA rearrangements induced by the knockout of *RECG* or *RECA1* genes, respectively[27], reproduced the results obtained previously using DNA gel blots. These results demonstrate that our junction read cluster analysis yielded efficient and precise identification of organelle DNA rearrangements and recombinations.

**Overview of links in mutants' chloroplast genomes.** The junction read analyses revealed remarkably biased distributions of organelle DNA in the mutants, in terms of position and direction, along with several mutant-specific patterns (Fig. 5 and Supplementary Table 10). In the junction read analysis of cpDNA, the

WT showed some minor links corresponding to those detected in paired-end analysis (Fig. 5 and Supplementary Fig. 9), showing basal level cpDNA recombination, which is also detected in WT by quantitative PCR[27]. By contrast, *RECG* KO cpDNA has numerous links that were clustered in the large IR and the region between 30,000 and 65,000, and these clusters were mainly composed of links in the same direction (Fig. 5). Notably, the locations of link clusters in the *RECG* KO cpDNA were related with prominent alteration of cpDNA dosage: drastic alterations at pos. 42,000 and 60,000 are likely to be the consequence of strong links between these loci (Figs. 2 and 5). *RECA2* KO cpDNA contained more links than the WT but fewer than *RECG* KO cpDNA. Also, in *RECA2* KO cpDNA, the links were distributed similarly to those of the *RECG* KO cpDNA in terms of location and direction (Fig. 5 and Supplementary Fig. 12). In two independent lines of *RECG* and *RECA2* KO mutants, cpDNA exhibited very similar link patterns (Supplementary Fig. 11), indicating that most of these links are not artifacts. Analysis of links in coding and non-coding regions shows these links locate more in non-coding regions of outside of genes in *RECA2* and *RECG* KO mutants' cpDNA (Table 3). No significant increase in links was detected in the cpDNA of mitochondrial *RECA1* KO mutants (Supplementary Fig. 11).

The categorized junction reads revealed that the number of products from recombination between repeats (mhr) was low in the WT, but moderately and greatly elevated in *RECA2* and *RECG* KO mutants, respectively (Table 2). We anticipated a correlation between repeat length and read count of these mhr products, since previous studies on *P. patens* chloroplast showed higher and lower accumulation level of products from recombination between 63-bp repeats and between 13–34 bp repeats, respectively[27]. However, analysis of the relationship between the length of repeats involved in recombination and their read counts revealed no significant correlation in any mutant; most of the recombination products were mediated by short repeats (microhomology: <20 bp) (Supplementary Figs. 12 and 13 A). mhmr data showed biased accumulation of one type of recombination product for most of rearrangements in *RECA2* and *RECG* KO mutants (Supplementary Data 3), supporting biased links regarding the direction. The number of recombination products without any sequence similarity at the junction (unk) was higher in the *RECG* KO mutant at the clustering region between 30,000 and 60,000, but not in the *RECA2* KO chloroplasts (Table 2 and Supplementary Data 3). No mutant exhibited an apparent alteration in the number of insertions (ins) and deletions (del), which are mainly located in A/T cluster in cpDNA, or links indicating U-turn–like rearranged DNA molecules around palindromic sequence (pal) (Table 2 and Supplementary Data 3). The mutants, as well as the WT, contained only small numbers of the U-turn–rearranged DNA molecules (pal; Table 2), in contrast to the case in *A. thaliana*[25].

**Overview of links in mutant's mitochondrial genomes.** In the analysis of the *RECA1* KO mtDNA, the abundance of links was drastically elevated compared with WT, mainly in one region (pos. 102,000–60,000 through 0), and most of the links in two separate areas share in the same direction (Fig. 6). Links in the first half (pos. 102,000–30,000) and the other half (pos. 30,000–60,000) were connected in the clockwise and anticlockwise directions, respectively, and these clusters were linked to each other. Interestingly, this link bias was common between the two lines of KO mutants (Supplementary Fig. 11); however, each individual link was not always shared between the lines (Supplementary Data 3). *RECG* KO mtDNA also contained a higher number of links that were largely distinct from those of

**Table 3 Likelihood of mutations occurring outside of gene or CDS.**

Values are given as chloroplast / mitochondrion.

| | Category | WT | RECA1 KO-1 | RECA1 KO-2 | RECA2 KO-6 | RECA2 KO-7 | RECG KO-1 | RECG KO-5 |
|---|---|---|---|---|---|---|---|---|
| Gene | del | 7.07 / 8.66 | 7.09 / 7.93 | 6.28 / 8.04 | 7.45 / 3.25 | 6.69 / 5.05 | 6.93 / 9.62 | 7.97 / 7.14 |
| | ins | 1.90 / 0.54 | 1.88 / 0.97 | 3.25 / 2.34 | 1.66 / 1.44 | 2.91 / 2.16 | 1.80 / 4.01 | 2.36 / 2.60 |
| | pal | – / – | 1.70 / – | 5.27 / – | – / – | – / – | 4.74 / – | 3.01 / – |
| | mhr | 1.51 / – | 12.29 / 1.28 | 1.76 / 1.12 | 1.76 / – | 1.44 / – | 1.20 / 0.74 | 1.18 / 0.95 |
| | unk | – / – | – / – | – / 1.08 | – / – | – / – | 10.54 / 2.88 | 10.54 / – |
| CDS | del | 4.61 / 2.06 | 4.59 / 3.09 | 3.98 / 2.89 | 4.78 / 0.77 | 4.48 / 2.06 | 4.63 / 3.70 | 4.88 / 2.26 |
| | ins | 1.12 / 0.77 | 1.71 / 1.35 | 1.88 / 1.33 | 1.46 / 0.77 | 2.05 / 1.55 | 1.37 / 1.55 | 1.71 / 1.11 |
| | pal | – / – | 6.15 / – | 2.73 / – | – / – | – / – | 2.88 / – | 1.56 / – |
| | mhr | 2.39 / – | 5.27 / 2.77 | 12.29 / 2.09 | 3.60 / – | 2.86 / – | 4.10 / 1.79 | 5.01 / 2.96 |
| | unk | – / – | – / – | – / – | – / – | – / – | 5.46 / 3.09 | 5.46 / – |

The left number is for chloroplast and the right number is for mitochondrion genome. Numbers larger than 1.00 is for mitochondrion genome. Numbers larger than 1.00 is for more mutation inside of gene/CDS, and less than 1.00 is for more mutation outside, and less than 1.00 is for more mutation inside of gene/CDS. The numbers are normalized for the length of gene/intergenic or CDS/non-coding regions.

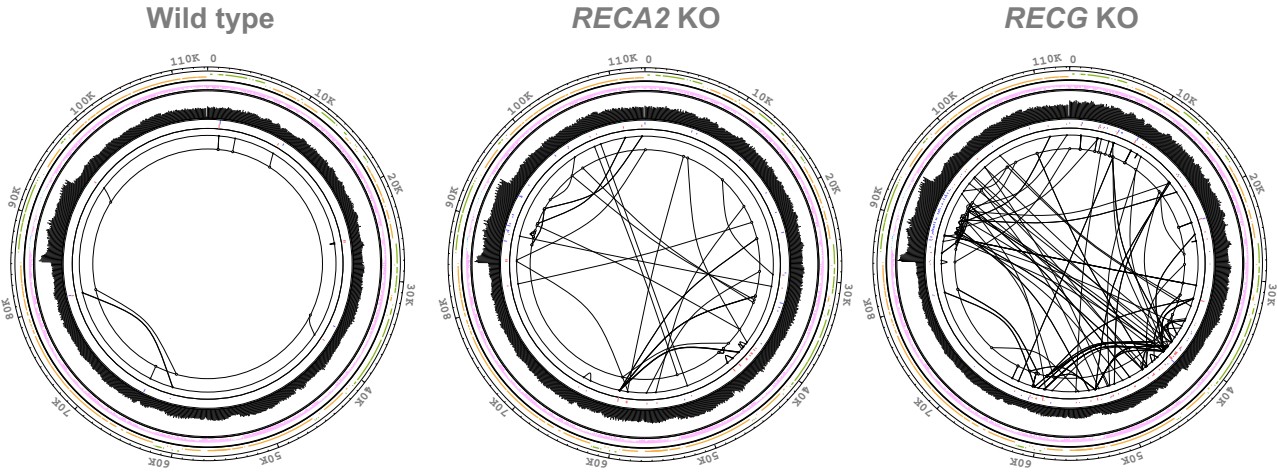

**Fig. 5 Links of junction reads in chloroplast DNA.** Rearrangements identified by junction reads are shown as links on the cpDNA map along with mapping depth. Detailed explanation of the tracks is found in Supplementary Fig. 8. One copy of the inverted repeats was removed.

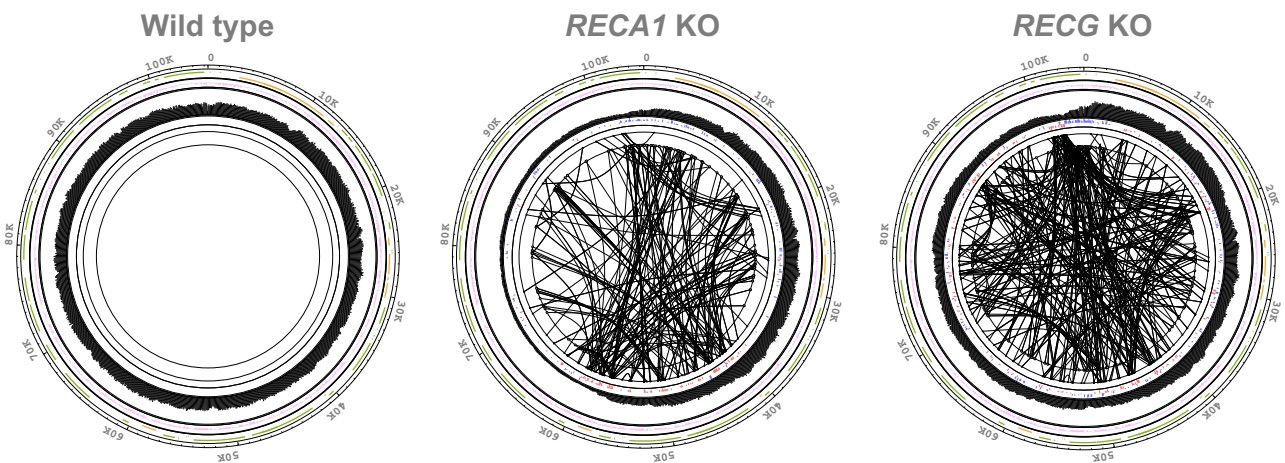

**Fig. 6 Links of junction reads in mitochondrial DNA.** Rearrangements identified by junction reads are shown as links on the mtDNA map along with mapping depth. Detailed explanation of the tracks is found in Supplementary Fig. 8.

*RECA1* mtDNA (Fig. 6 and Supplementary Fig. 12). *RECG* KO mtDNA links had a less biased distribution in terms of position (Supplementary Table 10), outside of a few hot spots. Among other genome regions, the region between 100,000 and 2000 had an extremely high density of links. This area can be further divided into two areas, 100,000–103,000 and 103,000–2000. The links between these areas were oriented in the clockwise and anti-clockwise directions, respectively, and were not connected to each other. Note that the read depth was significantly altered in these clustering regions. Analysis of links in coding and non-coding regions show most of these links locate more in noncoding regions outside of genes in *RECA1* and *RECG* KO mutants' mtDNA (Table 3). Knockout of chloroplast *RECA2* led to no significant increase in the number of links in mtDNA (Supplementary Fig. 11 and Supplementary Table 8), as expected.

Comparison of the number of the categorized links from junction reads revealed a significant increase of mhr (recombination with microhomology) links in *RECA1* KO mitochondria and an even greater increase in *RECG* KO (Table 2); thus most of the links observed in these mutants consist of products from recombination between SDRs. Whereas previous studies showed higher and lower accumulation of recombination products from longer (69–79 bp) and shorter (8–21 bp) repeats in *P. patens*

mitochondria[22,27], detailed analysis of the length of the repeats involved in mhr recombination and their read counts revealed no significant correlation, and most of the products were derived from recombination between repeats shorter than 20 bp (Supplementary Figs. 12 and 13B). As well as the case in chloroplast, mhmr data showed highly biased accumulation of one type of recombination products for most of the rearrangements (Supplementary Data 3). Knockout of *RECA1* or *RECG* caused a slight and a moderate increase in unk (recombination without sequence similarity) and ins/del in mitochondria, respectively (Table 2).

To analyze the correlation between links and repeats, we compared the distributions of links with those of repeats (Supplementary Fig. 12). The distribution of repeats in the mitochondrial genome exhibited no significant bias toward shorter (3–10 bp) repeats but some bias toward longer (≥10 bp) repeats (Supplementary Fig. 12). Neither *RECA1* KO nor *RECG* KO mitochondrial links exhibited a significant correlation with the bias of the distribution of repeats in the mitochondrial genome. Similarly, the distribution of repeats in the chloroplast genome exhibited a minor bias, but no significant correlation with the distribution of links, in the *RECA2* KO or *RECG* KO mutants (Supplementary Fig. 12). This suggested that most links of rearrangements distribute independent of repeat abundance.

## Discussion

In this study, we used ultra-deep sequencing and bioinformatics analysis to reveal that HRR mutations lead to dynamic alteration of organelle DNA. Deep and nearly homogeneously mapped reads indicate the occurrence of low-level heteroplasmic mutations from the overall organelle DNA. Our computational analysis, junction reads analysis and paired-end reads analysis, identified large number of mutations and recombinations other than point mutations all over the organelle DNA. Junction reads analysis, which have the advantage of determining mutations at sequence-level resolution, easily identified products from the recombination between 79 bp repeats, the longest dispersed repeats found in *P. patens* organelle DNA except for the chloroplast large IR, suggesting that our approach can detect recombination products in organelle DNA. In particular, previous analyses identified only small numbers of recombination products in mutants' cpDNA, but substantial numbers of recombination products in mutants' mtDNA[27]. Our approach identified numerous recombination products in the cpDNA of *RECG* KO mutants, comparable to those in mtDNA, with junction reads analysis (Table 2), again demonstrating the high sensitivity of our methods.

Our comprehensive and highly sensitive analyses yielded an overview of the mutations in organelle genomes in the HRR mutants, summarized as follows. First, mapping of the identified mutations, most of which consisted of products of recombination between SDRs shorter than 20 bp, revealed that the mutations were distributed in a highly biased manner in terms of their location on organelle genomes. Furthermore, the recombinations were biased in terms of their directions; one type of products for most of the SDRs were highly accumulated, implying recombinations are asymmetric. Accumulation of asymmetric recombination products is observed in *Arabidopsis* wild type and mutants as a hallmark of plant mitochondrial genome dynamics[32]. Our data also suggest that the directions of such asymmetry depend on their position in the mitochondrial genome. This tendency was proposed based on a rough analysis of the *A. thaliana MSH1* mutant mitochondrial genome focusing on specific repeats which shows asymmetry of recombination at neighboring repeats[32]. Moreover, such asymmetry in recombination are not obviously associated with local direction of transcription, just like what we have shown in *P. patens* mutants' organelle genomes. Although the length of repeats involved in these phenomena are different, >50 bp in *A. thaliana* and <50 bp in *P. patens*, these findings suggest locus-dependent asymmetry on recombination that are shared between *P. patens* and flowering plant mitochondria. Furthermore, we propose that such features are common in the chloroplast genome, as we have shown in our *P. patens* mutants. Previous data from our group and others suggested the involvement of specific repeats on mitochondrial genome instability[10,22,27,29]; in this study, however, most recombinations are found to occur with location preference but without specific sequence preference. Interestingly, many of the recombination products revealed by junction reads were not identical between the two mutant lines, but were localized in the same area, which is consistent with the other experimental observations. In organelle, studies on HRRs suggest their function in the repair of stalled or collapsed replication forks due to impediments on template DNA to suppress rearrangements[22,23,27]. Moreover, RECA and RECG are shown to be involved in the suppression in distinct pathways[28]. Recombination products as well as other mutations locate more in non-coding regions (Table 3), suggesting the mutations being less involved in transcription. Impediments other than transcription and the corresponding repair pathways may cause the mutant-specific bias of recombination.

Second, the number of recombination products was not significantly correlated with the length nor abundance of the repeats,

and most products resulted from recombination between repeats shorter than 20 bp, i.e., microhomology, while the mutations were localized in specific areas in each mutant. This could be attributed to the location-dependent accumulation of recombination products. The mapping results revealed that the difference in accumulated levels of previously identified mitochondrial recombination products, e.g., products of recombination between R10 (*nad2–atp9*) for *RECA1* KO and between R9 (*ccmF–atp9*) for *RECG* KO[27] are likely due to this biased localization of mutations, suggesting that RECA1 and RECG play different roles in organelle genome maintenance.

Third, our analysis revealed a relationship between some of these depth alterations and the recombination. Interestingly, read depth exhibited smooth alterations, with some minor exceptions, despite the accumulation of large numbers of recombination products in the organelle DNA of the mutants. This suggests that such recombination products are not transient but are integrated into the organelle genomes of plants (likely through DNA replication), thereby making organelle genomes into sub-genomes, as were shown in Marechal et al.[37] and Odahara et al.[22] for cpDNA and mtDNA, respectively. In combination with the depth variation that is often associated with asymmetric recombination, we propose a model for production of sub-genomic molecules with asymmetric recombination products via single DNA end-induced recombination between micro-homology, proposed as microhomology-mediated break-induced replication (Supplementary Fig. 15)[38]. In this model, a single DNA end induced by repair of a collapsed or stalled DNA replication fork recombines with an ectopic region with micro-homology to start replication, thereby generating subgenomes. Generation and retention of subgenomes that lack some part of the genome may result in depth variation. Interestingly, recombinations are taking place between clusters of locations rather than at random (Supplementary Fig. 11). This may be due to the three-dimensional orientation of folded genomic DNA, as in the bacterial chromosome[39,40], which has been largely unknown in chloroplasts or mitochondria so far.

In addition to recombination between SDRs, our analysis successively identified insertions and deletions in organelle DNA. Since most of the in/dels are located in A/T cluster in cpDNA, these mutations should be derived from replication slippage that is a major source of spontaneous mutation in chloroplast[41]. For mitochondria, we observe the number of insertion is increased in *RECA1* KO-1 and they are mainly insertion of A, T, or AT (Supplementary Data 3), suggesting the occurrence of replication slippage in mitochondria as well. Besides the higher copy number of cpDNA than mtDNA, such A/T cluster is more abundant in *P. patens* cpDNA than in mtDNA, and therefore, more del was detected in cpDNA. In the mutants, misannealing during recombinational repair may lead to increase of such indels.

Dosage variation along the mutants' organelle DNA, which was anticipated based on our DNA gel blot and quantitative PCR analyses[22,27], revealed that RECA and RECG play different roles in the maintenance of organelle genomes despite their common function in suppressing aberrant recombination between SDRs in mtDNA. The mapping of WT organelle DNA was almost flat, without any apparent slope (Supplementary Fig. 2), suggesting that organelle DNA replicates primarily from a non-specific origin via a mode that corresponds to recombination-dependent replication or rolling-circle replication[7]. Notably, the normalized mapping of the *RECA2* KO mutant's cpDNA (Fig. 2 C1 and C2) exhibited reduced depth and two peaks with mild slopes. Similarly, the normalized mapping of the *RECA1* KO mutant's mtDNA (Fig. 2 M1 and M2) showed slopes with a peak at pos. 35,000. These peaks imply cryptic cpDNA and mtDNA replication from a few loci that act as origins when the cell lacks HRR

activity. On the other hand, the normalized mapping results of the *RECG* KO mutant's cpDNA and mtDNA (Fig. 2 C3 and C4), which in particular showed somewhat increased depth with a slope, appears completely different from those of the *RECA1* and *RECA2* KO mutants' organelle DNA, providing further evidence for a difference in the roles of RECA and RECG in the maintenance of organelle genomes. Interestingly, the depth gap between the IR and LSC regions of the *RECG* KO mutant's cpDNA suggests depth of the two copies of the IR region are not identical (Supplementary Fig. 14).

In summary, we established a reliable and highly sensitive method to identify genome rearrangements by deep sequencing with short read. This revealed highly biased distribution of organelle genome rearrangements associated with production of various kinds of subgenomes. An alternative approach with long-read sequencing would give useful information to know both rearrangements and the detailed structure of such subgenomes, in the future study. Highly biased distribution of rearrangements suggests structural heterogeneity of organelle genomes, which should be related to the complex structure and replication of organelle DNA, both of which remain largely unknown. An integrative understanding of these mechanisms with HRR would solve the maintenance and dynamics of organelle genomes governed by HRR.

## Methods

**Plant materials and growth conditions**. *Physcomitrella patens* subsp. *patens* was used in this study. Knockout mutants of *RECA1*, *RECA2*, and *RECG* were generated by the polyethylene glycol (PEG)-mediated protoplast transformation method using linearized vectors pMAK124, pINO25, and pMSD202 and subsequent selection with appropriate antibiotics[22,23,27]. Protonemal cells of these strains were cultivated at 25 °C for 4 days on BCDATG medium[42] after homogenization in a blender.

**Extraction of genomic DNA**. *P. patens* total genomic DNA was extracted from protonemal cells by the CTAB method[43]. Homogenized frozen protonemal cells were suspended in CTAB buffer (2% CTAB, 1.4 M NaCl, 100 mM Tris-HCl pH8.0, 20 mM EDTA), and then genomic DNA was extracted by chloroform. The genomic DNA was precipitated by decreasing NaCl concentration and then dissolved in TE. After treatment with RNaseA to remove RNA, the genomic DNA was purified by ethanol precipitation and again purified in 0.75 M NaCl and 7.5% PEG.

**Library preparation and sequencing for Illumina**. Genomic DNA from *P. patens* was fragmented to ~300 bp average length on a Covaris S220 (Covaris) with a micro TUBE AFA Fiber Snap-Cap (Covaris), and subsequently subjected to size selection by using SPRI beads (Beckman Coulter). Libraries were prepared from the fragmented DNA using TruSeq DNA PCR-free (Illumina). The libraries were sequenced with 150 bp paired-end reads on an Illumina HiSeq X (Illumina).

**Quantitative PCR analysis of organelle DNA**. Quantitative PCR analysis of organelle DNA was performed using *P. patens* total genomic DNA, primers, and PowerUp SYBR Green Master Mix (Thermo Fisher Scientific) on a QuantStudio 12 K Flex Real-Time PCR System (ThermoFisher Scientific) with a PCR cycling condition (1 cycle for 50 °C 2 min and 95 °C 2 min, and 40 cycles for 95 °C 15 s and 60 °C 1 min). The relative copy number of organelle DNA determined by ΔΔCt method was normalized against that of the nuclear gene encoding actin.

**Reference sequence**. As for the reference genome of the organelle of spreading earth moss (*Physcomitrella patens*) used in this study, we downloaded genome sequence of chloroplast (NC_005087.2: 122,890 bp) and mitochondrion (NC_007945.1: 105,340 bp) from NCBI. The genome of chloroplast has a pair of identical sequence regions as inverted repeat (85,212-94,800 and 113,302-122,890, 9,589 bp). Since we cannot distinguish these two regions when we map read sequences, we removed the latter part of the reference for the read mapping. Resequencing of *P. patens* chloroplast and mitochondrial DNA revealed that the WT strain that we used harbored several mutations in its organelle DNA relative to registered chloroplast [NC_005087.2] and mitochondrial genomes [NC_007945.1] (Supplementary Table 11). Therefore, we used modified sequences for the chloroplast [LC516510] and mitochondrial [LC516511] genomes. The modification includes two SNPs for mitochondrion, and 39 SNPs and 27 INDELs for chloroplast (Supplementary Table 11). The longest INDEL modified here was 3 bp long.

We downloaded Ppatens_318_v3.fa.gz from JGI[17] for the reference genome of the chromosomes of *Physcomitrella patens*. This file consists of 27 almost complete

scaffold for chromosome sequences including undetermined regions, and 351 shorter scaffold segments. The overall sequence length is approximately 479 million base pairs. To use these data as a reference sequence for read mapping, we connected all these 388 sequence segments into a single sequence putting 500 "N" characters between each sequence segments. We did not apply any correction of minor mutations for these chromosome sequences.

**Mapping**. We have been developing a program for NGS read mapping called mpsmap[35] and we used it in this study. This program uses index of fixed length for reference and searches the mapping position of reads with minimum mismatch. We set index length as 14 bps in this study. This algorithm cannot detect the appropriate mapping position if the index region includes the mismatch. To minimize this problem, we repeated the search shifting the position of index in read. It finds the best mapping position when the maximum number of mismatches per read is small, or the mismatch positions are concentrated on a specific part of the read. The latter is the case we expect for the mapping of junction read. In this study, we used two sets of mapping conditions. The first condition is allowing two mismatches as maximum for each 150 bps read and repeat the search three times shifting the index position (h2r3). The second condition allows 75 maximum mismatches per read and repeat the mapping ten times with shifting the index position (h75r10). The first condition guarantees the best matching position for reads less than two mismatches, while the second condition may not be able to locate the position with least mismatch, if mismatches are scattered across the read. Nevertheless, we find the latter condition (h75r10) works fine to detect appropriate mapping positions of most junction reads. All the junction reads suggesting rearrangements are later confirmed that both side of segments match perfectly with reference at corresponding positions. We also performed a mapping using reads truncated to 100 bps by cutting off 50 bps from tail of the read, to analyze the rearrangement using the paired-end sequencing information. Using these truncated reads, we performed mapping allowing two mismatches at most in 100 bps and repeated the search with three different index positions (chp100_h2r3). Read pairs of paired end sequencing were mapped independently, and the relative mapping positions were analyzed later. All the programs described here are available for download at (https://github.com/NGS-maps/maps_organelle_v1.1). Detailed usage of the programs along with the description of file format can also be found at the site.

**Identification of rearranged DNA by paired end sequencing**. By analyzing the mapping results of paired end sequencing, we can estimate the rearrangements occurring between paired reads (magenta line in Supplementary Fig. 4a). To maximize the number of rearrangements detected in the sequencing segments, we truncated each read to 100 bps (the condition described in the previous section chp100_h2r3). Without rearrangement, paired reads should be mapped facing to each other within the approximate segment length on the reference. If there is a rearrangement within the segment, the mapped positions and direction of paired reads are separated farther than the distance expected from the length of sequencing segment, or point to the same direction when mapped on the reference genome. Supplementary Fig. 4a shows three such examples. Black lines indicate the rearranged genome, and black arrows indicate the rearrange positions and directions. Blue and green lines indicate sequence segments and read pairs derived from each segment. In cases 1 and 3 of Supplementary Fig. 4a, mapped positions of paired reads are far apart, and in cases 2 and 3, mapped directions of paired reads are pointing the same direction. Using an inhouse program called midhr developed for the rearrangement analysis, we first collected read pairs meeting one of the following criteria. (1) Mapped position of paired reads are separated more than 600 bp. (2) Paired reads are pointing toward same direction on the mapped reference. Read pairs satisfying these criteria are output in a file (with suffix.pdist). Then these read pairs sharing nearby mapping positions are clustered. The clustering method is simple. The first pair becomes the seed of the first cluster. For the remaining pairs, if the sum of the difference of mapped positions of paired reads is less than 500 bps with the seed pair of an existing cluster, the pair is assigned to the cluster, and if there is no cluster that a pair satisfies the previous condition, it becomes the seed of a new cluster. A program called sort_pdist carries out this operation. The output file (with suffix.pe) includes the mapped position of seed pair reads, number of members, standard deviation of the mapped positions among members of the cluster. This output file is used as an input to draw arcs representing the observed rearrangements.

**Identification of rearranged DNA by junction reads/clusters**. In the following procedure, detection of rearrangements is performed using the 150 bps read data, and paired reads are initially treated as two independent sequences. We used the mapping results of immap allowing many mismatches (condition h75r10 described above) for this analysis. Then we used our program midhr for the following analysis as well. First, mapped reads with more than three mismatches, and all of those mismatches are localized on one side of the read (i.e., there is no mismatches on the other side), are collected as junction read candidates. For each of these reads, the first occurrence of the mismatch when traced from mismatch-free end of a read, is identified as a junction point. In Supplementary Fig. 4b, the green arrow represents a mapped read, and the orange part of the arrow is the mismatch prone

region of the read. The junction-point is at the boundary of these green and orange regions. Reads sharing this junction-point on reference sequence are then clustered into a group, called a junction cluster. Members of a junction-cluster may have different mapping positions, while their junction points share the same position on reference (Supplementary Fig. 5a). These junction-reads share the same sequence on the side that matches the reference, and the sequences on the other side of the junction point may or may not be identical among junction reads. If they share the same sequence within the majority of the junction cluster, it is most likely because they share the same origin. By locating the position of this consensus sequence, we should be able to identify the other end of the rearrangement on the reference sequence. The procedure to define this mismatch consensus sequence is depicted in Supplementary Fig. 5b. First, we align the mismatch sequences of reads in a junction cluster, starting from the junction point. Then choose the most abundant base as a temporary consensus at each position. When the most common base at a site is not unique, we temporarily chose one according to the preference order of ATGC. For instance, if T and G are most common with same number of appearances, we choose T as consensus base at the position for the moment. The temporary consensus sequence is truncated at the position where the number of junction reads that matches the consensus sequence is less than a threshold value (3 by default). We then compared members of the junction cluster with this temporary consensus and removed members that do not match with the consensus (light gray sequence in Supplementary Fig. 5b). Construction of the final consensus sequence was made using remaining reads (orange sequences in Supplementary Fig. 5b) and then is truncated at the point where the number of matching reads becomes <4. The resulting consensus sequence is searched on the reference genome. After the matching position is found, all mismatch sequences of the junction reads are compared with the rearranged location and reads with more than one mismatch are removed from the junction cluster. Then we selected junction clusters that imply a genome rearrangement with the following three criteria. (1) The final consensus sequence is longer than 15 bps. (2) The number of remaining members (that match the temporary consensus) is more than three and is more than half of the junction reads of the original junction cluster. (3) The consensus sequence can be mapped on a unique position of reference genome. Rearrangements detected by this procedure include relatively short INDELs, rearrangements associated with homology/microhomology, rearrangements that do not accompany any homologous sequences, and U-turn like rearrangements. Here the insertion includes only tandem repeat and not the insertion of arbitrary sequences. They are classified and output in separate files (with suffix, .ins, .del, .mhr, .unk., and pal.) for the following analysis including the data visualization. The.ins and.del files include relatively short insertions or deletions. They are observed as rearrangements in the same direction and we classified this kind of rearrangements shorter than 50 bps as INDEL. Rearrangements with distant rearrange points are classified as rearrangement associated with homology/ microhomology (.mhr) if the rearrangement point shares the repeat sequence longer than 2 bps and classified as rearrangement with unknown mechanism (.unk) if there is no appropriate repeat sequence. There have been reports for U-turn like rearrangement with which the rearrangement points are nearby and in the opposite direction[25]. This kind of rearrangement appears to be caused by palindrome/quasi-palindrome sequences whose ends are homologous as reverse compliment (Supplementary Fig. 7). We classified rearrangements associated with homology/ microhomology (.mhr) whose rearrangement points are within 70 bps and opposite direction as a U-turn-like rearrangement and output to a file with suffix (.pal).

### Unification of rearrangements associated with homology/microhomology (.mhmr). 
This procedure unifies rearrangements associated with homology/ microhomology (.mhr) caused by identical repeats into one entry. As is shown in Supplementary Fig. 6, a set of repeat can cause two kinds of rearrangements (Rearrangement 1 and Rearrangement 2), and each rearrangement can be detected as two junction clusters (a and b). Therefore four .mhr entries can be related to a pair of repeat. After generating .mhr entries, all repeats associated with rearrangements are organized. Then all junction reads are re-examined to see if they are likely to be involved in the rearrangement caused by those repeats. With this procedure, the number of reads included in the .mhmr file is more than .mhr file, in general. Generation of .mhmr file is carried out by the program midhr along with files for other rearrangements, and no extra procedure is required.

### Comparison of rearrangements among wild type and mutant strains. 
Some of the rearrangements detected by the procedures described above may be shared with other WT/mutant strains. To organize the rearrangements, we made small programs (comp_ins, comp_del, comp_mhr, comp_mhmr, comp_unk, comp_pal, and comp_pe). Rearrangements sharing the same positions on reference are considered to be identical, and the numbers of reads indicating the rearrangement are output as a CSV (comma-separated values) format file (.out), and the read count scaled with the number of reads mapped in each experiment (.out2).

### Visualization of detected rearrangements. 
To visualize the positions of observed rearrangements, detected by both paired-end analysis and junction cluster analysis, we developed a program circmap that generates postscript image that resembles CIRCOS[44]. Supplementary Fig. 8 shows a part of image generated by circmap. The figure consists of several tracks of concentric circles. The scale on outside indicates the genome position along the circle. Two most inside tracks with filled and empty small circles on them represent rearrangement points, and arcs inward connecting these points represent the links corresponding to each rearrangement. The thickness of the arcs represents the number of reads (log scaled) implying the rearrangement. Filled and empty circles indicate the direction of the sequence from the rearrangement points. For instance, filled circles indicate the rearrangement occurs toward the clockwise direction, and the rearrangement is toward the counter clockwise direction at the empty circle. The third track from inside indicates the GC contents of the reference with 100 bps windows. The fourth track indicates gene regions with blue (clockwise direction) and red (counter clockwise direction) arcs. The fifth track indicates the depth of read mapping with 100 bps windows. The outmost (sixth) track just inside of the scale shows the depth change expected from the rearrangement. When there is a rearrangement junction toward the clockwise direction (black circle) the read depth is expected to decrease along the clockwise direction. On the contrary, rearrangements toward the counterclockwise direction (white circle) is expected to increase the read depth along the clockwise direction. The degree of the depth change depends on the number of reads involved in the rearrangement. Therefore, number of reads for all rearrangements in a window is added and shown as a blue bar inward when the population is expected to increase and shown as a red bar outward when the population is expected to decrease along the clockwise direction.

**Reporting summary**. Further information on research design is available in the Nature Research Reporting Summary linked to this article.

**Statistics and reproducibility**. Statistical analysis except for calculating Morisita index have been carried out using R program package (version 3.4.1). Morisita index was calculated with our in house program (h2h_morisita) which is available at the github site shown in the code availability section. All experiments are based on two independent mutant lines.

## Data availability
The *P. patens* organelle genome project has been deposited at DDBJ under BioProject number PRJDB9164. Genome sequencing data were deposited in the DDBJ Sequence Read Archive (DRA) database under accession number DRA009451. Source data underlying Fig. 1 is presented in Supplementary Data 4. All other data are available from the corresponding authors upon reasonable request.

## Code availability
All the programs used in this paper, including the program to calculate Morisita index, are available for download at (https://github.com/NGS-maps/maps_organelle_v1.1). It includes 22 executables compiled under Mac OSX 10.15.6 together with their source codes. Detailed usage of the programs both in English and Japanese are available at the site.

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

## Acknowledgements

We are grateful to S. Watanabe and H. Yoshioka for technical advice on Illumina library preparation. This work was supported by SUMITOMO Foundation (170946 to M.O.), the Japan Society for the Promotion of Science (16K18588 and 19K22405 to M.O.), and the Strategic Research Foundation Grant-aided Project for Private Universities (S1201003 to Y.S.).

## Author contributions

M.O., K.N. and T.O. performed experiments, M.O., K.N., Y.S. and T.O. analyzed data, and M.O., K.N. and T.O. wrote the paper.

## Competing interests

The authors declare no competing interests.
