## [Peer Review File · Communications Biology]

Reviewers' comments:

Reviewer #1 (Remarks to the Author):

Brief summary of the manuscript

The study focuses on the rearrangements in the organellar genomes of *Physcomitrella patens* when three nuclear-encoded genes known to be involved in recombination and repair of cytosolic genomes are knocked-out individually. The rearrangements are analyzed by deep paired-end sequencing of each mutant and compared to the wild type, considering evidence from junction and paired-end reads.

Overall impression of the work

The work is of great interest, in particular to those working on plant biology, organelle genetics and genome recombination. I believe that the methodology is powerful, although it should be analyzed very carefully to avoid any artifactual results. Given that most rearrangements are inferred to take place at short repeats, it is key to rule out mapping problems in which reads mapped in the wrong location suggesting an arrangement that it is not true. Such relevant information demonstrating solid methodology that would increase confidence in the results should be present in the main text, clearly stated and in sufficient detail, instead of in the supplementary material.

Specific comments

1. Introduction: The goals of the study should be addressed more thoroughly, indicating that mutants will be prepared to evaluate Also it is not clear why *E. coli* is included in the study?
2. Line 41: Mitochondrial genomes in land plants can be as large as ~11 Mb in *Silene conica*.
3. Line 73: Is it correct that *AtRECA1* is targeted to the cp and *Physcomitrella RECA1* is targeted to the mt?
4. Line 140: why were reads mapped mainly with 50% mismatches allowed? Even though the mapping was also performed with 2 mismatches, all the analyses were based on the mapping allowing 75 bp of the 150bp-long reads. Given that the mtDNA and cpDNA of this particular line of *P. patens* was resequenced and used as reference, I can't understand why so many mismatches were allowed. Also, given the presence of so many short repeats involved in the recombination events observed in the mutants, the 50% mismatches allowed, could introduce artifacts as they can be mismapped generating the artifactual evidence of recombination across the correct placement of the read and the mismapped location. In fact, the authors state in M&M in the suppl text that "may not be able to locate the best mapping position". This point is highly relevant as the whole study is based on these data. Thus, it should be explained very clearly and in great detail. In fact, I would expect to choose a perfect mapping (no mismatches) for the analyses based on the pair reads. Why were mismatches expected when mapping the pair reads based on the same DNaseq used to modify the reference genomes?
5. Also, I don't understand the .ins and .del files. In the mutant lines, did you observe <50bp indels? If so, which mechanism would explain this indels?
6. Why in Fig S1, the mapping rate of Illumina reads do not reach 100%?
7. Line 142: The mentioned "little or no increase in RECA1 and RECG KO" is in comparison to the wild type?
8. Line 206: organelle DNA contains no repeats longer than 90bp. That is, apart from the IR in the cpDNA! The citations for these statements do not point to articles that describe the repeats in each organelle....or do they? This important information, the size and amount of repeats, should be clearly stated or represented in a figure.
9. Lines 217-281: add reference to Table 2.
10. Lines 219-220: I don't have access to Suppl Data 1-12.
11. Lines 227: "all read pairs in close vicinity were clustered in the paired-end analysis". Is it possible that these pairs arise from different recombination events in a particular genomic region? How do you distinguish different events from a more confidence of a particular event? I can't find information regarding the size of the clusters. How many unusual read pairs form part of each clusters? I see part of this information in Table S4 but not sure about the range of the cluster sizes. Are there a few large

clusters and many clusters with a few members?

12. Lines 238-240: comparison from previous work: R9 and R10? Are these repeats?

13. Line 257-258: Even though RECA2 mutant shows more links than the WT, the increase is minimum in comparison to those in the RECG KO. It is also very different to the increase in links observed in the mtDNA for the RECA1 mutant. How do you explain this finding?

14. Line 293: I'm not sure why is the area 100,000 to 130,000 mentioned here as the mtDNA and cpDNA are smaller than 105kb and 122 kb, respectively.

15. Table 2, why so many "del" in cpDNA across WT and mutants? And why so few "del" in the mtDNA?

16. Lines 346-347: sentence very difficult to understand. What do you mean by "low level heteroplasmic mutations homogeneously from the organelle DNA"?

17. Line 350: junction reads determine mutations at the sequence-level? Where is this information? Did you find point mutations in the cpDNA or mtDNA?

18. Lines 353-355: this statement is hard to understand. Were the 79 and 47bp the longest repeats found in the mtDNA and cpDNA? Which "longer repeats" do you refer to as being harder to capture?

19. Lines 360-361: this study shows an overview of the mutations in organelle genomes of MUTANT individuals. It is important to state that when the nuclear-encoded genes involved in HRR are not knocked-out, the mutations (rearrangements) that you found do not take place (in the wild type).

20. Lines 363-364: "highly biased in terms of location and direction". By referring to a particular direction do you mean that they are asymmetric?

21. Lines 365-366: "asymmetric recombination is also observed in Arabidopsis and maize" The reference here is to a work in maize from 1995. Relevant references should be included. And also, please check if the asymmetric recombination were observed in MUTANTS or in wild type and modify the statement accordingly.

22. Lines 371-373: The previous work was also done in mutants of Physcomitrella and also involved recombination across short repeats?

23. Lines 374-382: I'd like to see a hypothesis explaining why there is a bias in the location of this short recombination events observed in the mutants, here and in previous studies. Even though they do not involve the same sequence, the authors mentioned that they are located in the same region/area. What is the length of the regions you refer to? How apart are the points of recombination within these areas of hotspots of recombination? How do you explain this bias? And why are these biases different in RECA and RECG mutants?

24. Lines 397-398 and Fig. S13: The model shows a single origin of replication, but this is certainly not true for plant organelle DNA and I would not include it in the model.

25. By single end-induced recombination, do you refer to MMBIR? Your explanation of the process fits with MMBIR and it would be good to mention this.

26. Line 426-429: The analyses and results done in E.coli are not relevant to the study or the relevance is not clearly stated. I suggest removing all the information on Ecoli WT and mutants as not providing significant findings and it is not particularly helpful to better understand the data on the moss. The manuscript is already too long and full of suppl tables and figures and would benefit from a reduction in number of figures.

27. General comment: When mutations are mentioned, it should be clarified that you refer to rearrangements/products of recombination and not to point mutations.

28. TableS8: If I understood correctly, this table shows the polymorphisms in the mt and cp genomes of the *P. patens* individuals published before and in this study. I would improve the title of the table. Also, I'm not sure about the SNPs in the mtDNA from "N" to T and "Y" to T. Were there N and Y in the published mtDNA of *P. patens*?

Reviewer #2 (Remarks to the Author):

The manuscript of Odahara et al. describes the organelle genome rearrangement in *Physcomitrella*

patens by comparing ultra-deep genome sequencing data of HRR mutants and wild type strain. They hypothesize that chloroplasts and mitochondria share a common mechanism for replication-related rearrangements based on comparative analysis of mtDNA and ptDNA in *P. patens* and expanded to *E. coli* strains to test their idea. Overall, the authors have provided several lines of evidences from their experiments and used in-house established bioinformatic pipeline for analyses, which seems to be sufficient for this journal. However, a lot of statements are lacking in detail and ambiguously written, and I still couldn't fully agree with their hypothesis from some point of view (see below).

1. Methodological issues

Authors selected Illumina paired-end sequencing to test the rearrangement in organelle genomes, however I do not agree with this idea. According to supplementary information (Supplemental Methods-Reference sequence), authors themselves have pointed out their weakness, "Since we cannot distinguish these two regions when we map read sequences, we removed the latter part of the reference for the read mapping.", which could distort the entire result of this paper. I think this is a key point authors need to describe further in order to justify their vulnerability. Although Illumina sequencing produces high accuracy reads and a large amount of reads and paired end information, it is not suitable for this study largely due to inverted repeats. The authors have given the details to overcome the weakness of short read sequencing by developing their own analysis pipeline, but I still find it difficult to agree with their explanations.

Authors need to state more details as to why they didn't used general mapping tools (e.g., bowtie2, bwa-mem) and develop their own methods. The statement about 'immap' does not give enough clues to use this. Authors also need to state about the mapping details, how they treat about multiple loci mapped reads, unpaired reads, both directional mapped reads (paired) but do not fit the insert size selection (Illumina library), etc... According to Fig S4, it is not clear that junction reads seem to be unpaired read, but how they treat with paired end reads that a single strand read has partial mismatches, but another strand has full matches. This is one example, but there are many cases to be described more details.

Overall, I would strongly suggest providing long-read sequencing data to verify authors' argument. It might overcome the inverted repeat problem, and more importantly, provide direct evidence of mutational recombination products. Some long-read based studies have been reported in plant, but I couldn't find any methodological advantages of short-read based work compared to previous study (e.g., Ruhlman et al., 2017, Am J Bot). If the long-read mapping result is still consistent, it can be an alternative and more cost-effective way to provide re-arrangement verification experiments using the authors' method.

2. Statistical test: statement about "significance"

I found many of 'difference' and 'similar' arguments are given without any statistical evidence. For example,

In Line 307-316 (Figure S12): the authors used "no significant bias", "a significant correlation with" without any clues, which cannot be determined only by the opinion of the authors. The authors should give a statistical test (e.g., use regression plot to show a link between repeat and connection).

In Line 426-429, the authors use significant differences, without any statistical statement, which seems to be the favored speculative idea for authors.

In Line 290 (Figure 6), the paper deals with the difference between the position and direction of the relation as a result of a recombination, but it would be useful to demonstrate clearly what the difference between position and direction implies.

Similarly, as I read this manuscript, there are a few other sentences that I could easily found ambiguity. It would be improved the manuscript if the authors provide statistical tests to support the significance of the observed trends.

3. Figure 2, Figure S3: considering chloroplasts, mitochondria numbers

If a mutant has different copies of chloroplast, mutant's cpDNA quantity could be differ from wild type strain not because of over-replication. Authors should clearly justify and describe about the relationship of correlation between mapping coverage and over-replication at the beginning of paper

that is not affected by another factor. According to previous studies (e.g., Fig 3 in Odahara et al., 2015, PLoS Genetics), it seems to have different numbers of chloroplast per cell in HRR mutants compared WT strain. Also, there is amplification-bias in Illumina sequencing platform. Therefore, I am not quite sure that standardization would work well for the data.

4. Recombination models in organelle genome

According to Line 409-413, the authors have stated "organelle DNA replicates primarily from a non-specific origin via a mode that corresponds to recombination-dependent replication", however, this does not fit to the model in Figure S13, which is hypothesizing to have bidirectional replication system. Also, from previous studies and this study, they might share some recombination mechanisms, however mapping result of mutant show that they don't share all the replication system. One way to test the replication system in chloroplast and mitochondria can be the GC-skew analysis, which could give information about replication origin or replication terminal. I would recommend a rewriting of the discussion section.

Other comments:

1. Supplementary Information: "As for the reference genome of the organelle of spreading earth moss (*Physcomitrella patens*) used in this study, we downloaded genome sequence of chloroplast (NC_005087.1: 122,890bp) and mitochondrion (NC_007945.1: 105,340bp) from NCBI. The genome of chloroplast1 has a pair of identical sequence regions as inverted repeat (85,212-94,800 and 113,302-122,890, 9,589bp)."

Why authors have used 'NC_005087.1', although corrected version 'NC_005087.2' have been released? 'NC_005087.2' has sequence variations (3 variants: atpB gene, chlB gene) and annotations are updated (3' & 5' rps12 genes) in version 2.

2. Line 335-337: In Figure 7B, there is no description of the intermediate peak in recG mutant. The authors should explain more details about this peak.

3. Line 292-293: This sentence is not about the genome of mitochondria (mtDNA genome size: 105 kbp), so it should be removed.

4. Supplementary Table 1: Q20 (99% accuracy) is meaningless for the outcome of Illumina sequencing, and Q30 (99.9% accuracy) is more commonly used to validate Illumina sequencing results.

5. Add more detailed explain about the abbreviations; 'hmr', 'pal', 'unk', 'ins', 'del' in the Table 2.

6. It will be better to include the discussion about the coding region and non-coding region.

7. Supplementary Information: "Nevertheless, we find these conditions work fine to detect appropriate mapping positions of most junction reads.": need explanations about why, give evidences about this.

Response to reviewers' comments

We would like to thank the reviewers for their valuable comments. In response to the reviewers' comments, we have carried out additional experiments and analysis, and revised our manuscript. We believe that the revised version of the manuscript is now sufficient for publication in *Communications Biology*.

Reviewers' comments:

Reviewer #1 (Remarks to the Author):

Brief summary of the manuscript

*The study focuses on the rearrangements in the organellar genomes of *Physcomitrella patens* when three nuclear-encoded genes known to be involved in recombination and repair of cytosolic genomes are knocked-out individually. The rearrangements are analyzed by deep paired-end sequencing of each mutant and compared to the wild type, considering evidence from junction and paired-end reads.*

Overall impression of the work

The work is of great interest, in particular to those working on plant biology, organelle genetics and genome recombination. I believe that the methodology is powerful, although it should be analyzed very carefully to avoid any artifactual results. Given that most rearrangements are inferred to take place at short repeats, it is key to rule out mapping problems in which reads mapped in the wrong location suggesting rearrangement that it is not true. Such relevant information demonstrating solid methodology that would increase confidence in the results should be present in the main text, clearly stated and in sufficient detail, instead of in the supplementary material.

Response) We described our methods in more detail in Results and updated Figure 3 (originally Figure 1) to make it clearer. To confirm the reliability of our junction cluster analysis, we calculated the overall matching ratio of the junction reads which is more than 99.9%, when each segment of the read is placed at appropriate locations (Table S4). The mapping position of paired reads for these junction reads are also confirmed for all cases. We believe the data and detailed method provided are sufficient to demonstrate our methodology being solid.

Specific comments

1. Introduction: The goals of the study should be address more thoroughly, indicating that mutants will be prepared to evaluate Also it is not clear why is *E.coli* included in the study?

Response) We have addressed the goal of our study and added the aim of the *E. coli* study as follows.

Revised: p.6) **In order to understand the role of HRR in the maintenance of organelle genomes**, we performed deep sequencing analysis of cpDNA and mtDNA on the Illumina NGS platform to comprehensively and thoroughly identify the genomic rearrangements. Using the Illumina NGS data, we followed two independent approaches to detect the rearrangements, split read mapping and paired-read mapping on the genome. We included *E. coli* HRR mutants' genomes to these analyses **to confirm that the phenotypes are specific for organelle**.

2. Line 41: Mitochondrial genomes in land plants can be as large as ~11 Mb in *Silene conica*.

Response) We have included the size of *Silene conica* mitochondrial genome with citation in the introduction.

Revised: p.3) **Land plant mtDNA, which is also mapped as a circular structure, varies in size from 50 kb to 3 Mb, but can be as large as 11 Mb in *Silene conica*¹.**

3. Line73: Is it correct that *AtRECA1* is targeted to the cp and *Physcomitrella RECA1* is targeted to the mt?

Response) Yes, correct. Since gene names in different species are confusing, we have noted about that in the manuscript.

Revised: p.4) *A. thaliana* has three RECAs, which are localized to the chloroplast (*AtRECA1*), chloroplast and mitochondria (*AtRECA2*), and mitochondria (*AtRECA3*)⁹. **Note that gene name and localization of their products do not correspond between *P. patens* and *A. thaliana* regarding *RECA* genes¹².**

4. Line 140: why were reads mapped mainly with 50% mismatches allowed? Even though the mapping was also performed with with 2 mitsmatches, all the analyses were based on the mapping allowing 75 bp of the 150bp-long reads. Given that the mtDNA and cpDNA of this particular line of *P. patens* was resequenced and used as reference, I can't understand

why so many mismatches were allowed. Also, given the presence of so many short repeats involved in the recombination events observed in the mutants, the 50% mismatches allowed, could introduce artifacts as they can be mismapped generating the artifactual evidence of recombination across the correct placement of the read and the mismapped location. In fact, the authors state in M&M in the suppl text that “may not be able to locate the best mapping position”. This point is highly relevant as the whole study is based on these data. Thus, it should be explained very clearly and in great detail. In fact, I would expect to choose a perfect mapping (no mismatches) for the analyses based the pair reads . Why were mismatches expected when mapping the pair reads based on the same DNaseq used to modify the reference genomes?

Response) The reason why reads were mapped with allowing 50 % mismatch is as follows. A junction read cannot be mapped on a single location on reference, because the former(left) and the latter(right) parts of the 150 bp read belong to different locations. Therefore, first we tolerated many mismatches to map each read, so that they can be mapped on the position where the longer part of the read matches. Then we located the position where the other part of the junction read matches. We explained the method in more detail in the article, and improved Fig. 3 to make this point clearer. The third column (Junction total) of Table S6 shows the overall mismatch ratio for each end of junction reads mapped on appropriate positions being less than 0.01% for all experiments.

We allowed maximum of two mismatches per read for the paired read analysis. The reason why we used both paired read analysis and junction read analysis, is their complementarity. In general, paired read analysis can detect more rearrangements (Table S5 and S8), while junction read analysis has higher resolution in terms of rearrangement position. This point is also mentioned in the revised article.

The sentence “may not be able to locate the best mapping position” refers to the difficulty shared by every mapping program using index method including bwa and other standard mapping tools. The significance of the problem is practically negligible, and we modified the sentence to avoid confusions.

Revised:

p.10) Computational detection of rearrangements from Illumina reads

To detect genomic rearrangements from the Illumina NGS data, we followed two independent approaches: paired-read mapping on the genome (Fig. 3a), and split (junction) read (Fig. 3b) methods (Fig. S4). Paired-end mapping, which used information from paired-end reads, has been used previously to detect the structural variants of the genome when paired reads are arranged in an unusual manner³⁴. In the paired end analysis, we truncated each pair reads from 150 bp to 50 bp (Fig. 3a).

Because we carry out rather stringent mapping for the paired end analysis, reads cannot be mapped if they include the rearrangement within the sequence. Truncation of reads allows the method to detect rearrangement occurring in the pale green region of Fig. 3a, i.e. 200 bp wider detection range. After mapping these truncated reads, we picked up read pairs separated farther than expected from the average insert length, and read pairs pointing toward the same direction. These paired reads imply the presence of rearrangement between them. Then read pairs sharing mapped positions in close vicinity are collected to form a paired end read cluster (Fig. 3a'). Each paired end read cluster suggests a rearrangement. In the example, the direction of sequence does not change during the rearrangement, therefore the directions of paired reads are facing to each other. On the other hand, if the direction of sequence alter during the rearrangement, as is in the case of U-turn rearrangements described later (Fig. 3c-5), the paired reads will point toward the same direction on the reference.

We call reads including a rearrangement within the 150 bp stretch as “junction reads”. The blue part of the junction read matches with the reference at position A and the orange part matches at A', as shown in Fig. 3b. To locate these two mapping positions for each junction read, we utilized the mapping method we developed, that does not allow gapped alignment, and allows mismatches as many as half the base pairs per read, while it guarantees the assigned position is the position with least mismatches for the read on the reference. With this method, junction reads are mapped on the position where the longer end of the segment matches (Fig. 3b). Artificial mapping results such as those of contaminated sequences, can be distinguished from junction reads, as mismatches for such misalignments distribute evenly along the read sequence. After the mapping, reads with mismatch-free region on one side (blue), and mismatches-prone region on the other end (orange), are collected as junction read candidate. Junction read candidates sharing the same junction point on the reference are gathered to form a junction read cluster and the consensus sequence of the mismatch prone region is taken and the position where this consensus sequence matches is located as the rearrange position (Fig. 3b' and Fig. S5). To re-evaluate each junction read cluster, we excluded reads with more than one mismatch in the shorter matching segment. We also excluded junction reads when their paired read is not mapped on the same reference sequence, or not mapped on appropriate position relative to the positions of the junction read.

Junction clusters are then categorized into five groups (Fig. 3c). They include short INDELs because we do not allow gapped alignment while mapping. INDELs shorter than 50 bps are categorized as (1) Deletion (del) and (2) Insertion (ins). Note here we only detect tandem repeat for insertions, and not insertions of arbitrary sequence. Rearrangement between two distant locations on reference are categorized as (3) Homologous rearrangement (hr) if there found homologous sequences longer than 3 bps responsible for the rearrangement, and (4) Rearrangement without homology (unk), if there found no homologous sequence around the junction points. Depending on the direction of homologous sequences, hr can be further categorized as same or opposite direction (Fig. S6).

Homologous rearrangements within close vicinity (< 70 bps) in opposite direction are categorized as (5) U-turn like rearrangements (pal; Fig. S7). The homologous sequence for these rearrangements can be observed as a palindrome, or quasi palindrome sequence.

Number of the junction clusters and the junction reads are summarized in Table 2. And matching ratio of the longer and the shorter segments of junction reads and pair reads of the junction reads are summarized in Table S4. The overall match ratio of junction reads is more than 99.9% and matching ratio of paired reads or junction reads is more than 96%. Some of the paired reads may include rearrangements as same/different as the corresponding junction read, and that is the reason why the mapping ratio of the paired reads is slightly lower. Nevertheless, we consider the high matching ratio suggest rearrangements detected by our junction read method are reliable.

By the paired end analysis, we can only identify the presence and approximate position of rearrangement (Fig. 3a') and cannot identify short indels. Also, a paired end read cluster may include more than two independent rearrangements in close vicinity. However, paired end analysis is generally more sensitive than the junction read analysis, and we carried out the analysis in order to support the results of junction cluster method. We show number of reads in junction read clusters and in paired read clusters, for some typical rearrangements, in Table S5.

5. Also, I don't understand the .ins and .del files. In the mutant lines, did you observe <50bp indels? IF so, which mechanism would explain this indels?

Response) ins and del are insertions and deletions shorter than 50 bp. Here ins include only the insertion by tandem repeat. We added the description of these in the revised text shown in the previous query. We also re-analyzed junction reads and revised Table 2. Since most of the in/dels are located in A/T cluster in cpDNA, these mutations should be derived from sequencing error or replication slippage that is a major source of spontaneous mutation in chloroplast (Massouh *et al.*, Plant Cell, 2016). For mitochondria, we observe the number of 'ins' is increased in *RECA1* KO-1 and the ins are mainly insertion of A, T, or AT, suggesting similar replication slippage in mitochondria. In the mutants, misannealing during recombinational repair may lead to increase of such indels. Because this result is unexpected and needs validation, we would like to avoid referring more about the indel in this paper.

6. Why in FigS1, the mapping rate of Illumina reads do not reach 100%?

Response) Thank you for the query. Fig. S1A shows the mapping rate of 150 bp Illumina reads allowing maximum of 2 bp mismatch per read, and they are about 86 % for all strains (numbers for each specific case can be found in Table S2). The mapping rate of reads truncated to 50 bp allowing up to 2 bp mismatch per read is shown in Fig. S1B and they are about 97% for all strains. These figures are to show the ratio of reads mapped to mt, cp and nc are similar with both mapping method, and to justify the estimate of mapping depth made with the mapping of truncated reads. The mapping rate of 150 bp reads with allowing maximum of 75 bp mismatch are about 99.6%, and we used this mapping method to carry out the junction read analysis. The paired read analysis was made using the truncated read mapping. And the mapping method with low mapping rate (h2r3) for 150 bp reads was not used for the rearrangement analysis. The reliability of the results from mapping allowing 75 bp mismatch was already explained (comment 4).

Part of the unmapped reads by the stringent method (h2r3) includes the junction reads, and the large portion of the rest of the unmapped reads should be the reads including sequencing errors.

7. Line 142: The mentioned “little or no increase in RECA1 and RECG KO” is in comparison to the wild type?

Response) Yes. We modified the sentence.

Revised: p.8) The number of reads mapped for cpDNA revealed little or no increase in *RECA1* and *RECG* KO mutants **as compared to the WT** (Fig. 1 and Table S2)

8. Line 206: organelle DNA contains no repeats longer than 90bp. That is, apart from the IR in the cpDNA! The citations for these statements do not point to articles that describe the repeats in each organelle....or do they? This important information, the size and amount of repeats, should be clearly stated or represented in a figure.

Response) Thank you for the comment. Yes you are right. *P. patens* organelle DNA contains no repeats longer than 90 bp, except for the IR in the cpDNA. We added Table S7 and Supp Data 1 and 2 showing organelle repeats longer than 15 bp. We hope this information will help readers to understand our manuscript in combination with the number and position of repeats shown in Figure S14.

Revised: p.14) **For junction read analysis, a 150 bp read can identify the recombination sites in most cases, because *P. patens* organelle DNA contains no repeats longer than 90 bp except for the large IR in cpDNA^{21,22} (Table S7 and Supp Data 1 and 2).**

9. Lines 217-281: add reference to Table 2.

Response) We cited Table 2 appropriately.

Revised: p.17) No mutant exhibited a apparent alteration in the number of insertions (ins) and deletions (del), or links indicating U-turn-like rearranged DNA molecules around palindromic sequence (pal) (Table 2). The mutants, like the WT, contained only small numbers of the U-turn-rearranged DNA molecules (Table 2), in contrast to the case in *A. thaliana*²³.

10. Lines 219-220: I don't have access to Suppl Data 1-12.

Response) We are sorry about the inconvenience. We uploaded the revised Supp Data 3-16 and should they be accessible now.

11. Lines 227: "all read pairs in close vicinity were clustered in the paired-end analysis". Is it possible that these pairs arise from different recombination events in a particular genomic region? How do you distinguish different events from a more confidence of a particular event? I can't find information regarding the size of the clusters. How many unusual read pairs form part of each clusters? I see part of this information in Table S4 but not sure about the range of the cluster sizes. Are there a few large clusters and many clusters with a few members?

Response) Different recombination events can be clustered in our paired-end analysis, and these are indistinguishable, unlike our junction read method. Each cluster is defined to be consists of at least three read pairs. The number of clusters and reads consisting them are listed in Tables 2 and S6, and detailed data are shown in Supp Data 3 and 16. The average numbers of reads in cluster can be estimated from these numbers (4 to 90 for junction reads, and 10 to 50 for paired reads). Numbers of reads of clusters for some typical homologous rearrangements can be found in Table S5.

12. Lines 238-240: comparison from previous work: R9 and R10? Are these repeats?

Response) Yes. We identified repeats in *P. patens* mitochondrial DNA and named them as R1-R23 (Odahara et al., 2018, Plant Phys). We revised the part so that the readers can easily understand them to be repeats. Also, we added information about the R1-R23 in the Supp Data 2.

Revised: p.15) In particular, the number of junction read clusters corresponding to products of recombination between repeats R9 (*ccmF-atp9*) or R10 (*nad2-atp9*) (Supp Data 2), which are hallmarks of mtDNA rearrangements induced by the knockout of *RECG* or *RECA1* genes, respectively²⁵, reproduced results obtained previously using DNA gel blots.

13. Line 257-258: *Even though RECA2 mutant shows more links than the WT, the increase is minimum in comparison to those in the RECG KO. It is also very different to the increase in links observed in the mtDNA for the RECA1 mutant. How do you explain this finding?*

Response) We think this is an interesting result too. As was already suggested in Odahara *et al.*, (Plant J, 2015), the difference may reflect the role of RECA2 and RECG in suppression mechanism. On the other hand, difference of the effect of RECA knockout in cp and mt may be due to structure (including property of repeats) of cpDNA and mtDNA, also suggested in Odahara *et al.*, (Plant J, 2015).

14. Line 293: *I'm not sure why is the area 100,000 to 130,000 mentioned here as the mtDNA and cpDNA are smaller than 105kb and 122 kb, respectively.*

Response) Thank you for the comment. The 130,000 should be 103,000. Revised.

Revised: p.17) This area can be divided into two areas, 100,000–103,000 and 103,000–2,000.

15. Table 2, *why so many "del" in cpDNA across WT and mutants? And why so few "del" in the mtDNA?*

Response) We found that most of these "del" are located at A/T cluster, thus this may be due to spontaneous deletion and/or sequencing error at A/T cluster. Besides the higher copy number of cpDNA than mtDNA, such A/T cluster is more abundant in *P. patens* cpDNA than in mtDNA, resulting that the "del" was detected more in cpDNA.

16. Lines 346-347: *sentence very difficult to understand. What do you mean by "low level heteroplasmic mutations homogeneously from the organelle DNA"?*

Response) Thank you for the comment. We revised as follows.

Revised: p.21) **Deep and nearly homogeneously mapped reads suggest identification of low-level heteroplasmic mutations from the overall organelle DNA.**

17. Line 350: *junction reads determine mutations at the sequence-level? Where is this information? Did you find point mutations in the cpDNA or mtDNA?*

Response) Sequence level information of junction reads is shown as Supp Data 3-16. It refers to the fact that junction analysis can spot the exact location of the rearrangement, while paired end analysis can only identify the presence of rearrangement in certain area. We haven't analyzed point mutations in the organelle DNA in this study as such mutations cannot be detected by our methods. We would like to analyze point mutations in organelle DNA as future study.

Revised: p.21) Junction reads analysis have the advantage of determining mutations at sequence-level resolution **as shown in junction read data**, whereas pair-ended reads analysis have the advantage of higher sensitivity.

18. Lines 353-355: *this statement is hard to understand. Were the 79 and 47bp the longest repeats found in the mtDNA and cpDNA? Which “longer repeats” do you refer to as being harder to capture?*

Response) We meant that the 79 bp and 47 bp repeats are the longest in the mtDNA and cpDNA, respectively, except for the large IR in the cpDNA. We revised the sentence as follows.

Revised: p.21) Junction reads analysis easily identified products from the recombination between 79 bp repeats, **the longest repeats found in *P. patens* organelle DNA except for the chloroplast large IR**, suggesting that our approach covered a large proportion of recombination products in organelle DNA.

19. Lines 360-361: *this study shows and overview of the mutations in organelle genomes of MUTANT individuals. It is important to state that when the nuclear-encoded genes involved in HRR are not knocked-out, the mutations (rearrangements) that you found do not take place (in the wild type).*

Response) We revised the sentence as follows, to note that the mutations are caused in the mutants.

Revised: p.21) Our comprehensive and highly sensitive analyses yielded an overview of the mutations in organelle genomes **in the HRR mutants**, summarized as follows.

20. Lines 363-364: *“highly biased in terms of location and direction”. By referring to a particular direction do you mean that they are asymmetric?*

Response) Yes, indeed this is included in the title of this manuscript. To facilitate readers understanding this, we have revised manuscript as follows

Revised: p.21) First, mapping of the identified mutations, most of which consisted of products of recombination between SDRs shorter than 20 bp, revealed that the mutations were distributed in a highly biased manner in terms of their **direction, which means asymmetric, and** location on organelle genomes.

21. Lines 365-366: *“asymmetric recombination is also observed in Arabidopsis and maize” The reference here is to a work in maize from 1995. Relevant references should be*

included. And also, please check if the asymmetric recombination were observed in MUTANTS or in wild type and modify the statement accordingly.

Response) We added a reference for Arabidopsis and revised the manuscript.

Revised: p.22) Accumulation of asymmetric recombination products is observed in maize **wild type** and *Arabidopsis mutants* as a hallmark of plant mitochondrial genome dynamics ^{8, 38}.

22. Lines 371-373: The previous work was also done in mutants of Physcomitrella and also involved recombination across short repeats?

Response) Yes, but our previous work was not comprehensive and focused only on recombination between relatively long repeats (>10 bp) analyzed by DNA gel blot or quantitative PCR analyses.

23. Lines 374-382: I'd like to see a hypothesis explaining why there is a biased in the location of this short recombination events observed in the mutants, here and in previous studies. Even though they do not involve the same sequence, the authors mentioned that they are located in the same region/area. What is the length of the regions you refer to? How apart are the points of recombination within these areas of hotspots of recombination? How do you explain this bias? And why are these biases different in RECA and RECG mutants?

Response) In response to another reviewer' s comment, we analyzed relationship between these recombination and transcription. The result showed that transcription is unlikely to participate in the recombination. Combining the results in this paper and the model of rearrangements in the HRR mutants proposed in previous papers, we think the difference of the biases might be related to replication stall or collapse. As explained in our previous paper, HRRs are functioning in the repair of stalled or collapsed replication fork due to replication impediments to suppress rearrangements. Furthermore, epistatic analysis showed RECA and RECG function in distinct pathways in suppression of rearrangements. Such aberrant recombination should be caused by impediments on template DNA. We think the difference in types of impediments and corresponding repair pathways which RECA and RECG are involved may cause bias and difference in the bias between RECA and RECG mutants. Regions that we refer to regarding the recombination is more than 30 kb in case of RECA1 mutant's mitochondria (pos.0-30,000 vs.32,000-60,000), and the distance between the areas are 50 kb at most (as shown in Figure 6). 3D structure of organelle nucleoid may be the clue to understand this distance as mentioned in the discussion.

Revised: p.22) In organelle, studies on HRRs suggest their function in the repair of stalled or collapsed replication forks due to impediments on template DNA to suppress rearrangements.^{21,22,25} Moreover, RECA and RECG are shown to be involved in the suppression in distinct pathways²⁶. Recombination products as well as other mutations locate more in non-coding regions (Table 3), suggesting the mutations being less involved in transcription. Impediments other than transcription and the corresponding repair pathways may cause the mutant-specific bias of recombination.

24. Lines 397-398 and Fig. S13: *The model shows a single origin of replication, but this is certainly not true for plant organelle DNA and I would not include it in the model.*

Response) Yes, you are right, we recognize that organelle DNA replication is not single origin. We propose a new model without a single replication origin to explain depth variations observed in the mutants (Fig. S14).

25. By single end-induced recombination, do you refer to MMBIR? Your explanation of the process fits with MMBIR and it would good to mention this.

Response) Thank you for your advice. We referred our model as MMBIR.

Revised: p.23) In combination with the depth variation that is often associated with asymmetric recombination, we propose a model for production of subgenomic molecules with asymmetric recombination products via single DNA end-induced recombination **between microhomology, proposed as microhomology-mediated break-induced replication (MMBIR)** (Fig. S14)⁴⁰. **In this model, a single DNA end induced by repair of a collapsed or stalled DNA replication fork recombines with an ectopic region with microhomology to start replication, thereby generating subgenomes.**

26. Line 426-429: *The analyses and results done in E.coli are not relevant to the study or the relevance is not clearly stated. I suggest removing all the information on Ecoli WT and mutants as not providing significant findings and it is not particularly helpful to better understand the data on the moss. The manuscript is already too long and full of suppl tables and figures and would benefit from a reduction in number of figures.*

Response) The goal of this study is to reveal overview of organelle rearrangements and to show that the rearrangement is “organelle-specific phenomenon”. Thus, we think comparing results with non-organelle *E. coli* with the same method is essential for this study. We believe this comparison will give insight into understanding of genome maintenance which would have impact for broad range of readers of Communications Biology.

27. *General comment: When mutations are mentioned, it should be clarified that you refer to rearrangements/products of recombination and not to point mutations.*

Response) We have paraphrased "mutation" to "recombination products" or "mutations except for point mutations".

Revised: p.21) Our computational analysis, junction reads analysis and paired-end reads analysis, identified large number of mutations **other than point mutations** all over the organelle DNA. Junction reads analysis have the advantage of determining mutations at sequence-level resolution as shown in junction read data, whereas pair-ended reads analysis have the advantage of higher sensitivity. Junction reads analysis easily identified products from the recombination between 79 bp repeats, **the longest repeats found in *P. patens* organelle DNA except for the chloroplast large IR**, suggesting that our approach covered a large proportion of **recombination products** in organelle DNA. In particular, previous analyses identified only small numbers of **recombination products** in mutants' cpDNA, but substantial numbers of **recombination products** in mutants' mtDNA²⁵. Our approach identified numerous **recombination products** in the cpDNA of *RECG* KO mutants, comparable to those in mtDNA, even with junction reads analysis (Table 2), again demonstrating the high sensitivity of our methods.

28. *TableS8: If I understood correctly, this table shows the polymorphisms in the mt and cp genomes of the *P. patens* individuals published before and in this study. I would improve the title of the table. Also, I'm not sure about the SNPs in the mtDNA from "N" to T and "Y" to T. Were there N and Y in the published mtDNA of *P. patens*?*

Response) We modified the title of the Table S13 (originally Table S8). Yes, the reference sequence for mitochondrial genome of *P. patens* (NC_007945.1) do include N and Y.

Revised: Table S13) List of mutations found and modified between the reference downloaded from the public database and our strain. The database ID of reference sequences of *Physcomitrella patens*, are NC_007945.1 for mitochondrion, and NC_005087.1 for chloroplast.

Reviewer #2 (Remarks to the Author):

*The manuscript of Odahara et al. describes the organelle genome rearrangement in *Physcomitrella patens* by comparing ultra-deep genome sequencing data of HRR mutants and wild type strain. They hypothesize that chloroplasts and mitochondria share a common mechanism for replication-related rearrangements based on comparative analysis of*

mtDNA and ptDNA in P. patens and expanded to E. coli strains to test their idea. Overall, the authors have provided several lines of evidences from their experiments and used in-house established bioinformatic pipeline for analyses, which seems to be sufficient for this journal. However, a lot of statements are lacking in detail and ambiguously written, and I still couldn't fully agree with their hypothesis from some point of view (see below).

1. Methodological issues

Authors selected Illumina paired-end sequencing to test the rearrangement in organelle genomes, however I do not agree with this idea. According to supplementary information (Supplemental Methods-Reference sequence), authors themselves have pointed out their weakness, "Since we cannot distinguish these two regions when we map read sequences, we removed the latter part of the reference for the read mapping.", which could distort the entire result of this paper. I think this is a key point authors need to describe further in order to justify their vulnerability. Although Illumina sequencing produces high accuracy reads and a large amount of reads and paired end information, it is not suitable for this study largely due to inverted repeats. The authors have given the details to overcome the weakness of short read sequencing by developing their own analysis pipeline, but I still find it difficult to agree with their explanations.

Authors need to state more details as to why they didn't used general mapping tools (e.g., bowtie2, bwa-mem) and develop their own methods. The statement about 'immap' does not give enough clues to use this.

Authors also need to state about the mapping details, how they treat about multiple loci mapped reads, unpaired reads, both directional mapped reads (paired) but do not fit the insert size selection (Illumina library), etc... According to Fig S4, it is not clear that junction reads seem to be unpaired read, but how they treat with paired end reads that a single strand read has partial mismatches, but another strand has full matches. This is one example, but there are many cases to be described more details.

Overall, I would strongly suggest providing long-read sequencing data to verify authors' argument. It might overcome the inverted repeat problem, and more importantly, provide direct evidence of mutational recombination products. Some long-read based studies have been reported in plant, but I couldn't find any methodological advantages of short-read based work compared to previous study (e.g., Ruhlman et al., 2017, Am J Bot). If the long-read mapping result is still consistent, it can be an alternative and more cost-effective way to provide re-arrangement verification experiments using the authors' method.

Response) Thank you for the comments. As explained in the Supplemental Method and

the Results (lines 150-152 of original version), we removed one of large inverted repeats (IR; 9.6 kb) from the cpDNA for mapping, just because reads assigned to these IRs cannot be distinguished. However, we do not think that this removal distorts our results. We consider the most important point of our analysis relies on the deep sequencing. The presence of only one junction read that suggest a rearrangement is not reliable enough. Here we require the presence of at least three junction reads that almost perfectly matches, to conclude a rearrangement. We agree the long-read analysis can provides different perspectives and planning to use them in the future study. However, we consider the long-read analysis has its own weakness such as the higher error-rate and the less coverage, and those are the reason why we decided to stick to the deep sequencing using Illumina NGS in this study. I hope you understand our point of view.

We described our methods in more detail in Results (subheading “Computational detection of rearrangements from Illumina reads”) and updated Figure 3 (originally Figure 1) to make it clearer. To confirm the reliability of our junction cluster analysis, we calculated the overall matching ratio of the junction reads which is more than 99.9%, when each segment of the read is placed at appropriate locations (Table S4). The mapping position of paired reads for these junction reads are also confirmed for all cases. We believe these improvements will help audiences to understand what was really carried out.

2. Statistical test: statement about ‘significance’

I found many of ‘difference’ and ‘similar’ arguments are given without any statistical evidence. For example,

In Line 307-316 (Figure S12): the authors used “no significant bias”, “a significant correlation with” without any clues, which cannot be determined only by the opinion of the authors. The authors should give a statistical test (e.g., use regression plot to show a link between repeat and connection).

In Line 426-429, the authors use significant differences, without any statistical statement, which seems to be the favored speculative idea for authors.

In Line 290 (Figure 6), the paper deals with the difference between the position and direction of the relation as a result of a recombination, but it would be useful to demonstrate clearly what the difference between position and direction implies.

Similarly, as I read this manuscript, there are a few other sentences that I could easily found ambiguity. It would be improved the manuscript if the authors provide statistical tests to support the significance of the observed trends.

Response) Thank you for the comment.

As for line 307-316, we calculated correlation coefficients between rearrangement link distributions (blue histogram) and corresponding repeat distributions (orange histogram) and wrote numbers in upper right of each plot with brown letters (Fig. S11). Most of the values are negative or very small (0.25 maximum), which support our conclusion, that there is no correlation between the link distribution and the repeat distribution.

As for line 426-429, we calculated correlation coefficient of mapping depth (Table S3. window size 10,000 for *E. coli* and 1,000 for organelles of *P. patens*) of each mutant compared to wild type to evaluate the degree of change caused by each mutation. Correlation coefficient of *E. coli recA* mutant with WT is 0.995, and that of *recG* mutant with WT is 0.889. Probably due to its characteristic abrupt increase of depth at the *ter* region, WT-*recG* mutant shows lower correlation coefficient, as compared to the other "unaffected" genomes such as chloroplast genomes of *RECA1* KO and mitochondrial genomes of *RECA2* KO.

As for line 290, comparing the *RECA1* KO and *RECG* KO charts of Fig. 6, it should be apparent that *RECG* KO has more links and some hot spots (i.e. 100k~0). However, we agree it is less obvious that *RECG* KO links has less biased distribution other than these hot spots. To evaluate the distribution bias, we calculated Morisita's I-delta index. As is shown in Table S10. I-delta indexes for *RECG* KO are generally lower than those of *RECA1* KO, which indicates the link distribution of *RECG* KO is generally more even.

3. Figure 2, Figure S3: considering chloroplasts, mitochondria numbers

If a mutant has different copies of chloroplast, mutant's cpDNA quantity could be differ from wild type strain not because of over-replication. Authors should clearly justify and describe about the relationship of correlation between mapping coverage and over-replication at the beginning of paper that is not affected by another factor. According to previous studies (e.g., Fig 3 in Odahara et al., 2015, PLoS Genetics), it seems to have different numbers of chloroplast per cell in HRR mutants compared WT strain. Also, there is amplification-bias in Illumina sequencing platform. Therefore, I am not quite sure that standardization would work well for the data.

Response) It is likely *RECA1* KO and *RECG* KO mutants possess different number of chloroplast and probably mitochondria. We defined "over-replication" for the *E. coli* mutants. But for *P. patens*, we defined "replication-related rearrangements" in the mutants by judging from the peaks and slopes observed in the mapping depth, and thus we believe the number of chloroplasts or mitochondria would not interfere this understanding. Because

of the Illumina bias, we utilized standardization to reveal the depth variation of the mutants. We believe our standardization is appropriate for evaluation of depth.

4. Recombination models in organelle genome

According to Line 409-413, the authors have stated “organelle DNA replicates primarily from a non-specific origin via a mode that corresponds to recombination-dependent replication”, however, this does not fit to the model in Figure S13, which is hypothesizing to have bidirectional replication system. Also, from previous studies and this study, they might share some recombination mechanisms, however mapping result of mutant show that they don't share all the replication system. One way to test the replication system in chloroplast and mitochondria can be the GC-skew analysis, which could give information about replication origin or replication terminal. I would recommend a rewriting of the discussion section.

Response) We assumed a single replication origin for mutant organelle to explain our results, however, the model may be uncommon to the organelle researcher and confuse general readers. We propose a new model without a single replication origin to explain depth variations observed in the mutants (Fig. S14). We also revised discussion section in order to explain recombination and replication model in the mutants as follows. GC-skew analysis was already tested for *P. patens* mitochondrial DNA (Terasawa, Odahara et al., Mol Biol Evol, 2007), but the results showed no obvious GC-skew bias supporting replication.

Revised:

p.22) **In organelle, studies on HRRs suggest their function in the repair of stalled or collapsed replication forks due to impediments on template DNA to suppress rearrangements.** ^{21, 22, 25} **Moreover, RECA and RECG are shown to be involved in the suppression in distinct pathways** ²⁶. **Recombination products as well as other mutations locate more in non-coding regions (Table 3), suggesting the mutations being less involved in transcription. Impediments other than transcription and the corresponding repair pathways may cause the mutant-specific bias of recombination.**

p.23) In combination with the depth variation that is often associated with asymmetric recombination, we propose a model for production of subgenomic molecules with asymmetric recombination products via single DNA end-induced recombination between microhomology, proposed as microhomology-mediated break-induced replication (MMBIR) (Fig. S14) ⁴⁰. **In this model, a single DNA end induced by repair of a collapsed or stalled DNA replication fork recombines with an ectopic region with microhomology to start replication, thereby generating subgenomes.** Generation and retention of subgenomes that lack some part of the genome may result

in depth variation. Interestingly, recombination are taking place between clusters of locations rather than at random (Fig. S9).

Other comments:

1. *Supplementary Information: “As for the reference genome of the organelle of spreading earth moss (*Physcomitrella patens*) used in this study, we downloaded genome sequence of chloroplast (NC_005087.1: 122,890bp) and mitochondrion (NC_007945.1: 105,340bp) from NCBI. The genome of chloroplast1 has a pair of identical sequence regions as inverted repeat (85,212-94,800 and 113,302-122,890, 9,589bp).”*

*Why authors have used ‘NC_005087.1’, although corrected version ‘NC_005087.2’ have been released? ‘NC_005087.2’ has sequence variations (3 variants: *atpB* gene, *chlB* gene) and annotations are updated (3’ & 5’ rps12 genes) in version 2.*

Response) Thank you for the information. We started the project four years ago and NC_005087.2 (Release date: 18 Jan 2019) was not available back then, and that was simply the reason why we used NC_005087.1. As we modified the reference sequence based on our WT sequencing, all three modifications from NC_005087.1 to NC_005087.2 (27,714 C>T, 57,788 G>A, 57,867 C>T) were included as listed in Table. S13. As for the nuclear genome, there was a major update in January 2018 (PhypaV3) and we used the new release.

2. *Line 335-337: In Figure 7B, there is no description of the intermediate peak in *recG* mutant. The authors should explain more details about this peak.*

Response) As described in the following sentence, this peak should be derived from over-replication at a locus where replication forks from opposite directions collide, as was reported in Rudlph et al. (Nature, 2013). We revised as follows.

Revised:

p.19) Relative to WT, mutants in *recA* exhibited no significant alteration of read depth profile, whereas mutation in *recG* led to a sharp increase in read depth in the terminus area and a reduced population around *oriC* (Fig. 7A), **indicating over-replication of sequences in the termination region due to replication fork collision**³⁶.

3. *Line 292-293: This sentence is not about the genome of mitochondria (mtDNA genome size: 105 kbp), so it should be removed.*

Response) 130,000 was a mistake. Corrected to 103,000.

Revised:

This area can be divided into two areas, 100,000–103,000 and 103,000–2,000.

4. Supplementary Table 1: Q20 (99% accuracy) is meaningless for the outcome of Illumina sequencing, and Q30 (99.9% accuracy) is more commonly used to validate Illumina sequencing results.

Response) We checked Q30 values of our Illumina sequencing data and replaced the Q30 values with the Q20 data in Table S1. The Q30 values are more than 87% and we believe these are sufficient for our informatics analysis.

5. Add more detailed explain about the abbreviations; ‘hmr’, ‘pal’, ‘unk’, ‘ins’, ‘del’ in the Table 2.

Response) We have added Fig. 3c and the detailed explanation for the types of the junction reads in Table 2, and added explanation in the main text.

6. It will be better to include the discussion about the coding region and non-coding region.

Response) Thank you for suggesting an interesting point. We analyzed rearrangements in coding and non-coding regions of organelle DNA (Table 3). For rearrangements by recombination between repeats (hr), all of the strains showed that the mutations are more frequent in non-coding regions suggesting the mutation being less involved in transcription. We added comments about the coding and non-coding region, in both the results and discussion sections.

Revised:

Results;

p.16) Analysis of links in coding and non-coding regions shows these links locate more in non-coding regions of outside of genes in *RECA2* and *RECG* KO mutants' cpDNA (Table 3).

p.18) Analysis of links in coding and non-coding regions show most of these links locate more in non-coding regions outside of genes in *RECA1* and *RECG* KO mutants' mtDNA (Table 3).

Discussion; p.22) Recombination products as well as other mutations locate more in non-coding regions (Table 3), suggesting the mutations being less involved in transcription.

7. Supplementary Information: “Nevertheless, we find these conditions work fine to detect appropriate mapping positions of most junction reads.”: need explanations about why, give

evidences about this.

Response) For the junction read analysis, we calculated the mapping ratio of junction reads for each end of the mapping positions (Table S4, the first and the second column). Overall matching ratio of junction reads (Table S4, the third column) is more than 99.9% for all cases, and corresponding paired reads are mapped on proximity of one of the mapping positions of the junction read in appropriate directions. These evidences can support the reliability of the rearrangements suggested by junction clusters. We revised main text accordingly as follows.

Revised:

p.13) Number of the junction clusters and the junction reads are summarized in Table 2. And matching ratio of the longer and the shorter segments of junction reads and pair reads of the junction reads are summarized in Table S4. The overall match ratio of junction reads is more than 99.9% and matching ratio of paired reads or junction reads is more than 96%. Some of the paired reads may include rearrangements as same/different as the corresponding junction read, and that is the reason why the mapping ratio of the paired reads is slightly lower. Nevertheless, we consider the high matching ratio suggest rearrangements detected by our junction read method are reliable.

Reviewers' comments:

Reviewer #1 (Remarks to the Author):

I thank the authors for responding each of the reviewers' queries and for providing the data sets for review. Based on the new information provided, I have several questions regarding the much better explained methods in the current version and on the interpretation of the results. In particular, I have serious concerns about the mapping strategy and interpretation of those results (see below). In addition, some of the responses to the reviewers' comments are either unclear or need to be revised. For example, I'm not satisfied with the rationale for including the *E. coli* experiments on the manuscript and in fact, I disagree with the goal and conclusions derived from the comparison of *E. coli* and *P. patens*.

The goal related to *E. coli* is unclear: "to confirm that the phenotypes are specific for organelle". First, the statement should have been specific to "plant" organelle. Second, the nuclear genome has not been inspected and thus, the effect of these RRR mutants could have an impact on the nuclear genome and thus, not be specific to organelles. Third, it is possible that other proteins in *E. coli* are at work reducing the effects of KO the specific RRR proteins tested in this work.

A thorough discussion of the *E. coli* results, previous studies and connection to the present work is missing.

Overall, I find that the manuscript is not well organized, the reader needs to look hard before finding the relevant information at any given paragraph. There is an excess of supplementary information (see below), which is not clearly explained or the reasoning for including it is not specified. Also, the figures and tables are missing key information to interpret them, either in the legends or footnotes. All suppl tables, figures and data should have all the necessary information to understand them. For example, all contractions (such as hmr, hr, in Table S2) should be spelled out in footnotes and the meaning of every color should be indicated.

The comments below follow the order of the manuscript:

lines 39-41: land plants are not always mapped as "a circular structure". There are plenty of examples in which the mtDNA could not be mapped either as a single master circle or due to repeats could not be mapped as a circle at all. In fact, *in vivo*, a single master circle has never been observed, at least in angiosperm mitochondria and instead they are found as branched, linear DNA molecules. Also, now another angiosperm holds the record: *Larix* at 11.7 Mb (Putintseva et al 2020 BMC Genomics)

lines 76-78: This sentence is not clear:

AtRECA2 or AtRECA3 is involved in the maintenance of mitochondrial genome stability by suppressing aberrant recombination between SDRs, while AtRECA3 has only minor suppressive effect 24.

lines 96-99: this statement could be improved for clarity:

"Despite extensive analyses, such phenomena have not been reported in *Escherichia coli* recA and recG mutants, suggesting that HRR factors play specific roles in plant organelles."

An involvement in genome instability has not been studied in *E. coli* or has not been reported because no evidence for a role in genome stability has been found for recA and recG in *E. coli*. If any studies have been done, please include references.

For example, Hong1995 (<https://www.embopress.org/doi/pdf/10.1002/j.1460-2075.1995.tb07233.x>) and others citing this article.

lines 102-104: There are several HRR proteins that have been mutated so I'm not sure which "HRR mutants" do you refer here as you then make comparisons to MutS mutants, which can also be considered HRR mutants. In addition, no reference to Arabidopsis HRR mutants are listed in the first statement.

Also the next statement: "This phenomenon is frequently observed as the consequence of recombination between intermediate-size repeats in angiosperm mitochondria 30"
Not sure what do you refer with "this phenomenon" given that asymmetric recombination is not mentioned in the cited reference #30. Indeed a reduction in recombination activity is described in that article. There is no mentioned of intermediate-size repeats or asymmetric recombination.
Given that the topic "asymmetric recombination" is included as a highlight of this study, even mentioning in the title, it should be analyzed clearly and provide solid evidence for this interpretation and also include a thorough comparison of previous studies. The manuscript is missing both and the findings and analysis presented are not sufficient to point to asymmetric recombination events". Even the meaning of asymmetric recombination is unclear as it may relate to different recombination mechanisms which are either not mentioned or mentioned in passing without an in-depth discussion.

lines 140-144: I'm not sure what do you mean by "strict" mapping when 2 mismatches are allowed. Are you considering errors in sequencing reads? Two mismatches out of 150bp reads (and even more out of 50 bp for truncated reads) is well above the sequencing error of Illumina sequencing (10–3 errors per base pair). What is the justification for allowing 2 mismatches in the mapping strategy? It seems unjustified. In fact, by allowing no mismatches, sequences errors could be discarded and only reliable mapped reads would be included. This may reduce the noise and in particular may show that wild-type plants do not show any rearrangements, as expected
How do you explain that mapping with 2bp mismatch yield 86% of mapped reads (FigS1A)? Why the other 15% of reads did not map in the wild-type?

Differences between FigS1 a and b indicate a reduction of read mapping across the three compartments (but the relative amount are similar). Does it mean that rearrangements occurred in all three compartments? I understand that the reduction of read mapping responds to unmapped reads due to rearrangements (junction reads) as mentioned by the authors in the response letter, and also sequencing errors. Would this explain the reduction in nuclear mapping too?

lines 160-165: why were two rounds of normalization were done?

lines 189-208: I find it very difficult to understand the reasoning for truncating the reads in the paired-end read analysis. At least as a comparison, I'd like to see this analysis without truncating the reads.

What is the difference between the paired-end read analysis, and the junction read? The pale green region is not mapping anywhere? What is the explanation for that?. The paired-end reads should map entirely (the 150 bp long read).

"Because we carry out rather stringent mapping for the paired end analysis, reads cannot be mapped if they include the rearrangement within the sequence."

If thee read includes a rearrangement, would that be considered a "junction read"? After spending a long time looking at Figure 3, I still can't figure out what and why did you follow this two strategies. In particular, the paired-end read analysis is very hard to follow.

Also, in the case 3a, a truncated read that maps a repeat, may yield artifactual results. Repeats should be masked out. That is, if a truncated read maps to a repeat, even non identical repeats (given the 2 mismatches allowed), they should be discarded. How do you treat these cases?

"Then read pairs sharing mapped positions in close vicinity are collected to form a paired end read cluster (Fig. 3a')."

Do you mean related to the same repeat? Or what would be the maximum distance between different repeat pairs to be considered a cluster? A read cluster has not been defined anywhere for paired-end reads.

lines 224-228: "We also excluded junction reads when their paired read is not mapped on the same

reference sequence, or not mapped on appropriate position relative to the positions of the junction read."

What would be appropriate position? Given that a rearrangement is detected, almost any position would be appropriate for the other read?

If the portion of the junction read with no mismatches is mapped fully within a repeat, this could be interpreted as a result of an alternative mapping and not of a rearrangement. How do you make sure this artefacts are not included? This would be a problem for the identical and non-identical repeats >75bp long.

lines 233-235: "Homologous rearrangement (hr) if there found homologous sequences longer than 3 bps responsible for the rearrangement,"

Only repeats longer than 20bp (at least) should be considered for homologous recombination, as it has been shown that RecA is only effective when two homologous regions are at least 20 bp long but its efficiency seriously increases with longer regions of homology. Below this length, non-homologous mechanisms are in play.

Table S5 shows several "typical" homologous sequences of <15 bp. These are NOT homologous sequences. They only share microhomology, which is not considered homologous.

Supp Data S4 also shows homologous sequences but most are very short...

Then, I suggest that hr and unk should be combined and only those repeats longer than 20 or 25 bp should be included in a separate category as hr.

Table S4 provides key information but there is sufficient missing information to prevent understanding it.

Table S5: CH in the table should be CP as in other figures.

Also, the second column shows "S" or "O" with no title. What do you mean here?

What do you mean by typical rearrangements? Are those examples or they are "typical" for any particular reason? Please explain.

lines 264-270:

Table S7 shows the identical repeats. It would be relevant for this study to include all repeats with >90% identity are present in each genome as these repeats are also considered homologous if longer than 20bp even when not identical, and they can recombine and would allow mismapping of reads giving the mismatches allowed.

A quick BLAST2seq that I ran indicated that there are none, but it would be better to make it explicit. And again, truncated reads mapping entirely over these repeats should be eliminated for the analysis to avoid artifactual results.

lines 272-275

Suppl Data 3 and 6:

I don't see the differences between Data S3 and S5 or between Data S4 and S6. They could be combined. For example in S3, the homologous regions involved in rearrangements (hr) are shown, while in S5 the repeats involved in rearrangements (hmr) are shown. And repeats and homologous regions are the same because the hr are identified as repeated regions. Also "hmr" is defined nowhere, and in particular is not mentioned in Fig. 3.

However, I identified a few cases where the results in Data S3 and S5 are not the same for the same repeat/hr. For example, row 141 in Data S5 and row 258 in Data S3 refer to the same repeats but the number of recombination events in each Data are not the same. Could you explain?

Also, a repeat (either in sense or reverse complement) is shown and counted 3 times in Dataset S3: rows 67, 246, 258. Please explain.

Also, why is Dataset S3 row 252, 150, or 131 not classified as indels instead of hr?

lines 282-284, the mentioned comparisons include only junction analysis but no comparisons with paired-end read analysis are presented.

lines 292-294: what do you mean by hallmarks of recombination? In which sense they stand out?

lines 304-305: what do you mean by "genome origin"? Do you refer to the origin of replication?

Figure 5:

What explanation exists for so many links in the WT paired end analysis? How many reads are conforming those clusters? It is expected to see that in chloroplast genome assemblies?

line 312-316: FigS10 should not be cited here as it is not relevant.

lines 320-322: Each of the recombination events (hr) in WT should be carefully examined as recombination across such short repeats are not expected in wild-type plants.

Figure S11.

I don't understand the histograms? Are the result of windows analyzed? what is the 6M mean?

line 326: "The number of recombination products without any homology at the junction (unk) was higher in the RECG KO mutant at hotspots for hr,".

Do you mean there are not even short repeats? None longer than 3bp?

What do you mean by "hotspots for hr? please clarify.

lines 346-347: what do you mean by "hot spots"?

The morisita index indicates if there is an even distribution or not. An uneven distribution does not imply the presence of hotspots.

E. coli analysis:

The E.coli sequence used as reference is the same strains sequenced for wild-type and the mutants? why wild type shows rearrangements?

A reference or plot of e. coli mapping pattern is missing.

Figure 7A using a different scale than the one mentioned in the text (position $\sim 3.9 \times 10^6$ bp and a minimum at position $\sim 1.6 \times 10^6$ bp) and it is confusing.

lines 385-386: "whereas mutation in recG led to a sharp increase in read depth in the terminus area and a reduced population around oriC (Fig. 7A),"

I see the opposite.

Discussion:

lines 424: I don't see evidence for asymmetric recombination products in the article on maize.

line 426: what do you mean by crude analysis?

lines 442-444: "Second, the number of recombination products was not significantly correlated with the length of the repeats (Fig. S10),"

Given that the maximum length of *P. patens* repeats is 79-90 bp and the fact that previously observed differences in recombination products are seen between large and intermediate-short repeats, what are the authors expecting from this analysis? Why would you expect any differences in repeats between 3 and 90bp? It might be more interesting to compare repeats below 20bp (nonhomologous)

and those above 25bp, which could be considered homologous.

Table S13; I would simply include the comparisons with the updated version of the cpDNA NC_005087.2 and in the revised version not even mentioned the "history" of first comparing with NC_005087.1 as it was the only available. Now, the version 2 is available and comparisons to the updated version should be included without mentioning the older version.

Finally, some of the responses were helpful to understand the reviewers' concerns but were not incorporated into the manuscripts. As many readers may have the same concerns, they should be included. Here are the two examples but please revise and also make sure you modify the text when responding to the reviewers.

The following statement provided to the reviewers should be incorporated in the text as no discussion of this topic is present in the current version:

"5. Also, I don't understand the .ins and .del files. In the mutant lines, did you observe <50bp indels? IF so, which mechanism would explain this indels?

Response) ins and del are insertions and deletions shorter than 50 bp. Here ins include only the insertion by tandem repeat. We added the description of these in the revised text shown in the previous query. We also re-analyzed junction reads and revised Table 2. Since most of the in/dels are located in A/T cluster in cpDNA, these mutations should be derived from sequencing error or replication slippage that is a major source of spontaneous mutation in chloroplast (Massouh et al., Plant Cell, 2016). For mitochondria, we observe the number of 'ins' is increased in RECA1 KO-1 and the ins are mainly insertion of A, T, or AT, suggesting similar replication slippage in mitochondria. In the mutants, misannealing during recombinational repair may lead to increase of such indels. Because this result is unexpected and needs validation, we would like to avoid referring more about the indel in this paper. "

The explanation for Fig.S1 could be included in the legend as it is likely that readers will have the same concerns.

"6. Why in FigS1, the mapping rate of Illumina reads do not reach 100%?

Response) Thank you for the query. Fig. S1A shows the mapping rate of 150 bp Illumina reads allowing maximum of 2 bp mismatch per read, and they are about 86 % for all strains (numbers for each specific case can be found in Table S2). The mapping rate of reads truncated to 50 bp allowing up to 2 bp mismatch per read is shown in Fig. S1B and they are about 97% for all strains. These figures are to show the ratio of reads mapped to mt, cp and nc are similar with both mapping method, and to justify the estimate of mapping depth made with the mapping of truncated reads. The mapping rate of 150 bp reads with allowing maximum of 75 bp mismatch are about 99.6%, and we used this mapping method to carry out the junction read analysis. The paired read analysis was made using the truncated read mapping. And the mapping method with low mapping rate (h2r3) for 150 bp reads was not used for the rearrangement analysis. Part of the unmapped reads by the stringent method (h2r3) includes the junction reads, and the large portion of the rest of the unmapped reads should be the reads including sequencing errors."

The following should also be incorporated in the discussion:

"15. Table 2, why so many "del" in cpDNA across WT and mutants? And why so few "del" in the mtDNA?

Response) We found that most of these "del" are located at A/T cluster, thus this may be due to spontaneous deletion and/or sequencing error at A/T cluster. Besides the higher copy number of cpDNA than mtDNA, such A/T cluster is more abundant in *P. patens* cpDNA than in mtDNA, resulting that the "del" was detected more in cpDNA."

Reviewer #2 (Remarks to the Author):

The authors have addressed most of my comments to this revised manuscript and I would like to reiterate that the results presented in this paper contribute to our better understanding of nature of organelle recombination. Together with the authors' thorough replies to the reviewer's comments, I am thinking that the present article is now ready for publication after minor revision as below.

1. I do understand author's point of view on long-read sequencing from their reply. However, I would suggest that the authors should add their opinions on the long-read sequencing data in the manuscript, because the authors' opinions will help other readers why authors used short-read sequencing without using long-read sequencing. This will help readers to know that there is another approach (long-read based) to recombination analysis.

2. Typo in Figure 3

- `map position ~' → `map position ~`
- `~ Paiered end ~' → `~ Paired end ~`
- `~ infered by' → `~ inferred by`

3. Figure legends in the main text and the figures do not match. Please correct it.

- (e.g.) Figure 1 in the main text: "Detection of rearrangement by (a) paired-end reads and (b) junction reads"; Figure 1 in the figures: "Ratio of mapped reads"

4. [Page 4, Line 73] The authors have stated "A. thaliana has three RECAs, which are localized to the chloroplast (AtRECA1), chloroplast and mitochondria (AtRECA2), and mitochondria (AtRECA3). Note that gene name and localization of their products do not correspond between P. patens and A. thaliana regarding RECA genes." in revised manuscript based on Reviewer 1's comment. However, determining correspondence between P. patens and A. thaliana RECA genes using only locality information isn't reasonable in my opinion. I would like to suggest a simple phylogenetic analysis to see a relationship between P. patens and A. thaliana based on RECA genes.

5. Reference format is not consistent. Please follow the format of Communications Biology.

6. I don't think it's necessary to place Figure 2 ("Ratio of mapped reads") in the main figure.

7. Supplementary Table 11: I would like to suggest changing Q20 (99% accuracy) to Q30 (99.9% accuracy) in this table as Supplementary Table 1.

Response to reviewers' comments

We would like to thank the reviewers for their valuable comments. In response to the reviewers' comments, we have carried out additional analysis, and revised our manuscript. Importantly, we have omitted *E. coli* data in the revised manuscript in response to the Reviewer#1's comments. In addition, we carried out paired-end analysis with 100 bp truncated reads to eliminate artifact, and the results have been shown as supportive data for junction read analysis. We believe that the revised version of the manuscript is now sufficient for publication in *Communications Biology*.

Reviewers' comments:

Reviewer #1 (Remarks to the Author):

I thank the authors for responding each of the reviewers' queries and for providing the data sets for review. Based on the new information provided, I have several questions regarding the much better explained methods in the current version and on the interpretation of the results. In particular, I have serious concerns about the mapping strategy and interpretation of those results (see below). In addition, some of the responses to the reviewers' comments are either unclear or need to be revised.

Comment 1) *For example, I'm not satisfied with the rationale for including the *E. coli* experiments on the manuscript and in fact, I disagree with the goal and conclusions derived from the comparison of *E.coli* and *P. patens*.*

*The goal related to *E. coli* is unclear: "to confirm that the phenotypes are specific for organelle". First, the statement should have been specific to "plant" organelle. Second, the nuclear genome has not been inspected and thus, the effect of these RRR mutants could have an impact on the nuclear genome and thus, not be specific to organelles. Third, it is possible that other proteins in *E.coli* are at work reducing the effects of KO the specific RRR proteins tested in this work.*

*A thorough discussion of the *E.coli* results, previous studies and connection to the present work is missing.*

Response) Thank you for your comments regarding *E. coli* data. We decided to omit *E. coli* data from our manuscript. We hope this would clarify the aim of this manuscript and give deeper understanding of organelle genome rearrangements.

Comment 2) *Overall, I find that the manuscript is not well organized, the reader needs to look hard before finding the relevant information at any given paragraph. There is an excess of supplementary information (see below), which is not clearly explained or the reasoning for including it is not specified. Also, the figures and tables are missing key information to interpret them, either in the legends or footnotes. All suppl tables, figures and data should have all the necessary information to understand them. For example,*

all contractions (such as hmr, hr, in Table S2) should be spelled out in footnotes and the meaning of every color should be indicated.

Response) We reconsidered all the figures and tables for their validity to be included in this manuscript, and gave them additional information as responses to the following comments.

The comments below are rearranged to follow the order in the manuscript:

Comment 3) *lines 39-41: land plants are not always mapped as “a circular structure”. There are plenty of examples in which the mtDNA could not be mapped either as a single master circle or due to repeats could not be mapped as a circle at all. In fact, in vivo, a single master circle has never been observed, at least in angiosperm mitochondria and instead they are found as branched, linear DNA molecules.*

Also, now another angiosperm holds the record: Larix at 11.7 Mb (Putintseva et al 2020 BMC Genomics)

Response) Thank you for the information about the plant mitochondrial DNA. Taking the information into account, we revised the sentence as follows.

Revised) p.3) Land plant mtDNA is also mapped as a circular structure and varies in size from 50 kb to 11.7 Mb ¹. They are, however, actually shown to form linear, branched, and circular structures in many plants ².

Comment 4) *lines 76-78: This sentence is not clear:*

AtRECA2 or AtRECA3 is involved in the maintenance of mitochondrial genome stability by suppressing aberrant recombination between SDRs, while AtRECA3 has only minor suppressive effect ²⁴.

Response) We revised the sentence as follows.

Revised) p.5) AtRECA2 or AtRECA3 is involved in the maintenance of mitochondrial genome stability by suppressing aberrant recombination between SDRs, while AtRECA3 has only minor involvement in the suppression ²⁶.

Comment 5) *lines 96-99: this statement could be improved for clarity:*

“Despite extensive analyses, such phenomena have not been reported in Escherichia coli recA and recG mutants, suggesting that HRR factors play specific roles in plant organelles.”

An involvement in genome instability has not been studied in E. coli or has not been reported because no evidence for a role in genome stability has been found for recA and recG in E. coli. If any studies have been done, please include references.

For example, Hong1995 (<https://www.embopress.org/doi/pdf/10.1002/j.1460-2075.1995.tb07233.x>) and others citing this article.

Response) We refer to the latter case. To our knowledge, no evidence for a role in suppressing recombination between short repeats has been found for *recA* or *recG*. In addition to a paper by Hong et

al., 1995 you suggested, such phenomena have not been reported in the mutants (Lloyd and Rudolph, 2016). According to the omission of *E. coli* data, the statement has been removed.

Comment 6) lines 102-104: *There are several HRR proteins that have been mutated so I'm not sure which "HRR mutants" do you refer here as you then make comparisons to MutS mutants, which can also be considered HRR mutants. In addition, no reference to Arabidopsis HRR mutants are listed in the first statement.*

Response) In this sentence, we referred only for *P. patens*. We revised as follows and cited Arabidopsis references accordingly.

Revised) p.6) Interestingly, *P. patens RECA1* and *RECG* mutants exhibit asymmetric accumulation of recombination products, which defines biased accumulation of one of the two types of reciprocal recombination products, in mitochondria^{22,27}, as in *A. thaliana* mutants of *MutS homolog 1 (MSH1)*³⁰, which also participates in maintenance of organelle genome stability by suppressing aberrant recombination between SDRs in both *A. thaliana*^{30,31} and *P. patens*²⁸.

Also the next statement: "This phenomenon is frequently observed as the consequence of recombination between intermediate-size repeats in angiosperm mitochondria 30"

Not sure what do you refer with "this phenomenon" given that asymmetric recombination is not mentioned in the cited reference #30. Indeed a reduction in recombination activity is described in that article. There is no mentioned of intermediate-size repeats or asymmetric recombination.

Response) We are sorry, the reference was wrong, should be Davila et al., 2011. We revised it.

Revised) p.6) This phenomenon is frequently observed as the consequence of recombination between intermediate-size repeats in angiosperm mitochondria³².

Ref 32. Davila JI, *et al.* Double-strand break repair processes drive evolution of the mitochondrial genome in Arabidopsis. *BMC Biol* **9**, 64 (2011).

Given that the topic "asymmetric recombination" is included as a highlight of this study, even mentioning in the title, it should be analyzed clearly and provide solid evidence for this interpretation and also include a thorough comparison of previous studies. The manuscript is missing both and the findings and analysis presented are not sufficient to point to asymmetric recombination events". Even the meaning of asymmetric recombination is unclear as it may relate to different recombination mechanisms which are either not mentioned or mentioned in passing without an in-depth discussion.

Response) Our definition of the asymmetric recombination/rearrangement is as follows. For each (micro-) homologous sequence on genome, there are two distinct rearrangements. Those are depicted in Fig. S6 as Rearrangement 1 and Rearrangement 2. When the frequency of these rearrangements are not the same, or only one of them is found, we call the rearrangement is asymmetric. mhm data (Supp. Data 7

and 8) shows this asymmetric recombination regarding most of repeats, and there are some examples in Table S5. There found only one of the rearrangements, for the second, fourth, fifth, sixth and eighth entry. Accordingly, we revised some parts in the manuscript as follows.

Revised)

[Introduction] **p.6)** Interestingly, *P. patens* *RECA1* and *RECG* mutants exhibit asymmetric accumulation of recombination products, which defines biased accumulation of one of the two types of reciprocal recombination products, in mitochondria^{22,27}, as in *A. thaliana* mutants of *MutS homolog 1 (MSH1)*³⁰, which also participates in maintenance of organelle genome stability by suppressing aberrant recombination between SDRs in both *A. thaliana*^{30,31} and *P. patens*²⁸.

[Discussion] **p.19)** Our comprehensive and highly sensitive analyses yielded an overview of the mutations in organelle genomes in the HRR mutants, summarized as follows. First, mapping of the identified mutations, most of which consisted of products of recombination between short dispersed repeats (SDRs) shorter than 20 bp, revealed that the mutations were distributed in a highly biased manner in terms of their location on organelle genomes. Furthermore, the recombinations were biased in terms of their directions; one type of products for most of the SDRs were highly accumulated, implying recombinations are asymmetric. Accumulation of asymmetric recombination products is observed in *Arabidopsis* wild type and mutants as a hallmark of plant mitochondrial genome dynamics³². Our data also suggest that the directions of such asymmetry depend on their position in the mitochondrial genome. This tendency was proposed based on a rough analysis of the *A. thaliana* *MSH1* mutant mitochondrial genome focusing on specific repeats which shows asymmetry of recombination at neighboring repeats³². Moreover, such asymmetry in recombination are not obviously associated with local direction of transcription, just like what we have shown in *P. patens* mutants' organelle genomes. Although the length of repeats involved in these phenomena are different, >50 bp in *A. thaliana* and <50 bp in *P. patens*, these findings suggest locus-dependent asymmetry on recombination that are shared between *P. patens* and flowering plant mitochondria.

Comment 7) lines 140-144: I'm not sure what do you mean by "strict" mapping when 2 mismatches are allowed.

Are you considering errors in sequencing reads? Two mismatches out of 150bp reads (and even more out of 50 bp for truncated reads) is well above the sequencing error of Illumina sequencing (10–3 errors per base pair). What is the justification for allowing 2 mismatches in the mapping strategy? It seems unjustified. In fact, by allowing no mismatches, sequences errors could be discarded and only reliable mapped reads would be included. This may reduce the noise and in particular may show that wild-type plants do not show any rearrangements, as expected

How do you explain that mapping with 2bp mismatch yield 86% of mapped reads (FigS1A)? Why the other

15% of reads did not map in the wild-type?

Response) With our understanding, given the error rate of illumina sequencing as 10^{-3} and the errors are randomly distributed, the chance of finding no error on a 150bp read is $(0.999)^{150} = 86\%$, and 14% of reads are expected to contain at least one error. Likewise, $(0.999)^{100} = 90\%$, $(0.999)^{50} = 95\%$ and therefore 10% of 100bp reads, and 5% of 50bp reads are expected to contain at least one error. Sup. Data. 1 and 2 contains the actual number of WT reads mapped with errors, on chloroplast, mitochondrion and nucleus. First they were mapped with allowing mismatches up to half the length of the read, and counted the number of mismatches for each read at their mapping position with the least mismatches. The table shows the number of reads, ratio, and cumulative ratio of reads with the mismatch numbers. The actual number of reads mapped without mismatches are about 5% less than the estimate made above. This is probably due to the fact that the sequencing quality is slightly lower than average, and also due to the fact that the distribution of the sequencing error is not even, but biased due to the sequence context of the genome. The latter explanation is consistent with our previous observations (ref. 33). This also explains the reason why mapping rate of 150bp reads allowing 2 mismatches stay 86 %. We inspected the mapping pattern (distribution of errors) of these reads carefully to confirm that most of these reads with mismatches are not misplaced.

Revised) p.7) Neither strict mapping nor mapping of the truncated paired-end reads (first 100 bp) changed the proportion of reads mapped to each genome (Fig. S1A and B). **Considering sequencing error, a maximum of 2 bp mismatches were allowed in each mapping, which improved mapping ratio substantially while ensuring mapping accuracy (Sup. Data 1 and 2).**

Comment 8) *Differences between FigS1 a and b indicate a reduction of read mapping across the three compartments (but the relative amounts are similar). Does it mean that rearrangements occurred in all three compartments? I understand that the reduction of read mapping responds to unmapped reads due to rearrangements (junction reads) as mentioned by the authors in the response letter, and also sequencing errors. Would this explain the reduction in nuclear mapping too?*

Response) As we explained in response to the previous comment 7, the decrease of the mapping ratio for longer reads are due to the sequencing errors, and they can be observed in mapping on nucleus. We changed the length of the truncated reads from 50bp to 100bp, with the reasons described later.

Comment 9) *lines 160-165: why were two rounds of normalization were done?*

Response) We apologize there were mistakes in our descriptions. We replaced the description as follows.

Revised) p.8) **To plot the net amount of organelle DNA, we divided the depth of mutants by the depth of WT at each window, and then normalized with the number of reads mapped on mutant and WT, for each organelle. This procedure resulted in a smooth and sensitive display of organelle DNA abundance (Fig. 2).**

Comment 10) lines 189-208: *I find it very difficult to understand the reasoning for truncating the reads in the paired-end read analysis. At least as a comparison, I'd like to see this analysis without truncating the reads.*

What is the difference between the paired-end read analysis, and the junction read? The pale green region is not mapping anywhere? What is the explanation for that?. The paired-end reads should map entirely (the 150 bp long read).

Response) Suppose there is a sequence segment of 400bp. Paired end sequencing reads 150bps from both ends of the 400bp leaving the 100bp in the middle unread. Junction reads method can detect rearrangement occurring in the middle of the one of the reads (pale green areas). On the other hands, the paired reads method can detect rearrangement in the 100bp unread region of the original sequence segment (light pink area). The reason why we truncate the read length for paired end analysis is to enlarge this unread region, or the area where we can detect the rearrangement. By truncating the reads to 100bp, the unread region becomes 200bp, and by truncating the reads to 50bp, the unread region becomes 300bp (magenta area). As such, read truncation can significantly increase the sensitivity of the paired end analysis. As you pointed out, read truncation down to 50bp can induce the erroneous mapping due to the sequence repeat longer than 50bp. We thought the artifact can be removed based on the positions of such long repeats. However, we failed to remove a couple of artifacts due to the flaw in our program. To eliminate this artifact, we replaced all the mapping analysis of truncated reads from 50bp to 100bp, instead of just fixing the flaw. This change eliminates the artifact caused by erroneous mapping, because the longest repeat other than the IR is 90bp. Although the number of detected rearrangements decreases by this change, the number of rearrangements for chloroplast is still more than junction read method, and we believe our paired end method provides a different perspective to the analysis. Paired end analysis with 150 bp reads would further reduce the number of rearrangements detected. Related to these, we revised Figure 5, 6, S9, S10, and S11.

Figure to explain the difference of Junction read method, and Paired read method, and the reason why reads are truncated for the paired end analysis.

Comment 11) “Because we carry out rather stringent mapping for the paired end analysis, reads cannot be mapped if they include the rearrangement within the sequence.”

If these read includes a rearrangement, would that be considered a “junction read”? After spending a long time looking at Figure 3, I still can’t figure out what and why did you follow this two strategies. In particular, the paired-end read analysis is very hard to follow.

Response) We carried out the two analysis (Junction reads and Paired reads) methods independently. Junction read method looks for reads with many mismatches on one side of the read when mapped on a single position of the reference. The mismatch prone end of the read can be mapped on the different part of the reference. This junction read includes rearrangement point inside, and the two parts of the read separated by the rearrangement point are mapped on different positions on reference. The pair read of a junction read is mapped within the distance expected from the length of the sequencing segment, from one of the two parts of the junction read.

On the other hands, the pair read method look for pair reads to be mapped with allowing only small number of mismatches expected from sequencing error. When the pair reads are mapped on distant positions further than expected from the length of sequencing segment, or pointing to the same direction, it suggests the presence of rearrangements between the pair reads.

A read in junction analysis includes the rearrangement point. Reads in paired end analysis do not.

Comment 12) Also, in the case 3a, a truncated read that maps a repeat, may yield artifactual results. Repeats should be masked out. That is, if a truncated read maps to a repeat, even non identical repeats

(given the 2 mismatches allowed), they should be discarded. How do you treat these cases?

Response) We apologize, in our previous analysis, we failed to exclude a couple of artifacts caused by repeat in reference. To eliminate such artifacts, we changed the length of truncated reads for paired end analysis from 50bp to 100bp. As we stated in the response to comment 10, we can eliminate the artifact by this change, because the longest repeat in reference other than IR is 90 bp.

Comment 13) *"Then read pairs sharing mapped positions in close vicinity are collected to form a paired end read cluster (Fig. 3a')."*

Do you mean related to the same repeat? Or what would be the maximum distance between different repeat pairs to be considered a cluster? A read cluster has not been defined anywhere for paired-end reads.

Response) Paired end reads are supposed to point to each other within the few hundred bps. When the paired reads are mapped on positions further than 1,000 bp, or mapped in the same direction, it suggests the presence of rearrangement between the pair reads. Read pairs sharing mapped positions in close vicinities may be the result of the identical rearrangement. These pair reads are clustered into groups. Read pairs are assigned to the cluster if sum of the difference of two read positions is less than 500bp with the seed pair of the cluster. If there are more than one rearrangement points sharing the same direction, read pairs from these rearrangements may be categorized into the same cluster. This is the limitation of paired end analysis, and the error can be avoided by junction cluster analysis. Details of the method are provided in " **Identification of rearranged DNA by paired end sequencing**" of the Supplementary Methods.

Comment 14) *lines 224-228: "We also excluded junction reads when their paired read is not mapped on the same reference sequence, or not mapped on appropriate position relative to the positions of the junction read."*

What would be appropriate position? Given that a rearrangement is detected, almost any position would be appropriate for the other read?

If the portion of the junction read with no mismatches is mapped fully within a repeat, this could be interpreted as a result of an alternative mapping and not of a rearrangement. How do you make sure this artefacts are not included? This would be a problem for the identical and non-identical repeats >75bp long.

Response) The former and latter parts of a junction read are mapped on different positions on reference sequence. The pair read of the junction read have to be mapped within the length of the sequencing segment (few hundred bps) from the latter part of the junction read. This is the appropriate position of the paired read of the junction read. If the mapped position of the paired read is not appropriate, the junction read is ignored. However, such a case was found to be very rare.

Comment 15) lines 233-235: “Homologous rearrangement (*hr*) if there found homologous sequences longer than 3 bps responsible for the rearrangement,”

Only repeats longer than 20bp (at least) should be considered for homologous recombination, as it has been shown that RecA is only effective when two homologous regions are at least 20 bp long but its efficiency seriously increases with longer regions of homology. Below this length, non-homologous mechanisms are in play.

Table S5 shows several “typical” homologous sequences of <15 bp. These are NOT homologous sequences. They only share microhomology, which is not considered homologous.

Supp Data S4 also shows homologous sequences but most are very short...

*Then, I suggest that *hr* and *unk* should be combined and only those repeats longer than 20 or 25 bp should be included in a separate category as *hr*.*

Response) Thank you for the comment, to avoid the confusion, we decided to call all the repeats causing rearrangements as micro-homology. It includes some sequence repeats longer than 20 bps. We did not distinguish them. Related to this, we revised Table S5 and Supp Data 5-8. According to this, *hr* and *hmr* are renamed to *mhr* and *mhmr*, respectively.

Comment 16) *Table S4 provides key information but there is sufficient missing information to prevent understanding it.*

Response) Table S4 shows the match ratio of junction reads mapped. In our analysis, the longer end of the junction read is allowed no mismatch and one mismatch is allowed for the shorter end. As a result, the matching ratio of junction reads is more than 99.7% for all mutants/WT. For the pair read of the junction read, there are some cases in which the rearrange point is also included in the pair read. We just checked the mapped position of the pair read as was described in 14, but did not check if they are junction read themselves, nor looked for the mapping position of the shorter segment if that is the case. Therefore, the matching ratio of the paired read is not as high as the junction read (> 96%). We added caption to Table S4.

Comment 17) *Table S5: CH in the table should be CP as in other figures.*

Also, the second column shows “S” or “O” with no title. What do you mean here?

What do you mean by typical rearrangements? Are those examples or they are “typical” for any particular reason? Please explain.

Response) All “CH” have been corrected to “CP”. “S” and “O” in the second column means direction of repeats explained in Fig. S6. We added description in Table S5. We showed them as examples of rearrangements to demonstrate correspondence between junction read clusters and paired read clusters.

Comment 18) lines 264-270:

Table S7 shows the identical repeats. It would be relevant for this study to include all repeats with >90% identity are present in each genome as these repeats are also considered homologous if longer than 20bp even when not identical, and they can recombine and would allow mismapping of reads giving the mismatches allowed.

A quick BLAST2seq that I ran indicated that there are none, but it would be better to make it explicit. And again, truncated reads mapping entirely over these repeats should be eliminated for the analysis to avoid artifactual results.

Response) In the present manuscript, we changed the length of the truncated reads to 100bp, and we believe all the artifacts caused by the repeat in reference sequence are eliminated. As of the concern about the artefact caused by less homology, we believe we are being able to avoid this with the following reasons. First, our mapping algorithm guarantees the mapped position of each read is the position with least mismatches. Second, if erroneous assignment of reads is caused by the tolerance of mismatches up to 2bp, the mismatches would occur at exactly the same location on the reference, and detected as a SNP, which is not found in our analysis. However, we agree the analysis on non-identical homology would be interesting and would like to consider it in the future study.

Comment 19) lines 272-275

Suppl Data 3 and 6:

I don't see the differences between Data S3 and S5 or between Data S4 and S6. They could be combined. For example in S3, the homologous regions involved in rearrangements (hr) are shown, while in S5 the repeats involved in rearrangements (hmr) are shown. And repeats and homologous regions are the same because the hr are identified as repeated regions. Also "hmr" is defined nowhere, and in particular is not mentioned in Fig. 3.

Response) Sup.Data 3 to 6 are now renumbered to 5 to 8. We are sorry about the confusion. Sup. Data 5 and 6 are hr, and 7 and 8 are hmr, and they are renamed to mhr and mhmr, respectively. They include redundant information. The mhmr data unifies the rearrangements caused by the same microhomology. Rearrangement 1 and Rearrangement 2 in Sup. Fig. 6 consists of two mhr entries (switching longer and shorter parts of junction read), so there are four mhr corresponding entries for one hmr entry, at most. Also, the numbers of reads included in mhmr can be more than those in hmr, because we recollected the reads based on the knowledge of micro-homology for mhmr. We added explanation for mhmr in the Supplementary method section and the main text as follows.

Revised) p.11) (3) Rearrangement based on microhomology/homology (mhr) if there found any identical sequences longer than 3 bps responsible for the rearrangement, and (4) Rearrangement without any homology (unk), if there found no homologous sequence around the junction points.

Depending on the direction of homologous sequences, mhr can be further categorized as same or opposite direction (Fig. S6). Because a pair of repeat can cause two sets of rearrangements (Fig. S6), we unified mhr data associated with the same pair of repeats as mhmr data, thereby number of two type of rearrangements associated with one pair of repeats can be compared (Fig. S6). Detailed procedure to form mhmr dataset is described in the Supp. methods.

Comment 20) *However, I identified a few cases where the results in Data S3 and S5 are not the same for the same repeat/hr. For example, row 141 in Data S5 and row 258 in Data S3 refer to the same repeats but the number of recombination events in each Data are not the same. Could you explain?*

Also, a repeat (either in sense or reverse complement) is shown and counted 3 times in Dataset S3: rows 67, 246, 258. Please explain.

Response) Entries for the mhr are selected according to the procedure to form junction read clusters described in Fig. S5. Entries for the mhmr are then re-selected based on the microhomology/homology found to be causing the rearrangements, and the procedure to count reads for each rearrangement is different. Therefore, the number of reads may be different. Numbers in new Supp. Data 5~8 are scaled with the number of reads for each mutants, and that is why it includes non-integer numbers. As was explained in the previous comment 19, maximum of four mhr entries (new Supp. Data 5,6) may correspond to one microhomology/homology. On the other hand, the number of mhmr entries (new Supp. Data 7,8) for one particular microhomology/homology is just one.

Comment 21) *Also, why is Dataset S3 row 252, 150, or 131 not classified as indels instead of hr?*

Response) Entries in ins and del are defined as indels shorter than 50bp. The line 252 entry of Sup. Data 3 is 98 bp apart, and line 131 entry is 54 bp apart. The line 150 entry is 151bp apart, and it is a U-turn rearrangement which is different from indels.

Comment 22) *lines 282-284, the mentioned comparisons include only junction analysis but no comparisons with paired-end read analysis are presented.*

Response) In this paragraph, we compared only data obtained by junction cluster with previous data obtained by DNA gel blot or quantitative PCR analysis. We revised it as follows.

Revised) p.14) *Next, we compared the data obtained by junction cluster analyses with data from previous organelle DNA analyses of RECA1, RECA2, and RECG KO mutants carried out by DNA gel blot or quantitative PCR.*

Comment 23) *lines 292-294: what do you mean by hallmarks of recombination? In which sense they stand out?*

Response) These recombination products are characteristically induced by knockout of RECG or RECA1. We rephrased “hallmark of” as “characteristic”.

Revised) p.14) In particular, the number of junction read clusters corresponding to products of recombination between repeats R9 (*ccmF-atp9*) or R10 (*nad2-atp9*) (Supp. Data 4), which are characteristic mtDNA rearrangements induced by the knockout of *RECG* or *RECA1* genes, respectively ²⁷, reproduced the results obtained previously using DNA gel blots.

Comment 24) lines 304-305: what do you mean by “genome origin”? Do you refer to the origin of replication?

Response) We meant position “0” as the “genome origin”. The phrase has been removed due to the update of results.

Comment 25) Figure 5:

What explanation exists for so many links in the WT paired end analysis? How many reads are conforming those clusters? It is expected to see that in chloroplast genome assemblies?

Response) As a result of re-analysis of paired end analysis with 100 bp truncated reads, WT organelles, especially chloroplast, still have some links. Some of these links in WT are corresponded to those in junction reads in WT and were detected by quantitative PCR in our previous works (Odahara et al., 2015, Plos Genet), and thus implying basal level recombination occurring in WT. Since these links are composed of clusters of less than 50 reads, as shown in Supp Data 17 and 18, such rearrangements are suggested to exist in cell in a very low-level, thus rarely affect genome assemblies. We added an explanation for the links in WT as follows.

Revised) p.15) In the junction read analysis of cpDNA, the WT showed some minor links corresponding to those detected in paired-end analysis (Fig. 5 and S9), showing basal level cpDNA recombination, which is also detected in WT by quantitative PCR ²⁷.

Comment 26) line 312-316: *FigS10 should not be cited here as it is not relevant.*

Response) Thank you for the comment. Fig. 5 should be cited here. We revised it.

Revised) p.15) Also, in *RECA2* KO cpDNA, the links were distributed similarly to those of the *RECG* KO cpDNA in terms of location and direction (Fig. 5 and S12). In two independent lines of *RECG* and *RECA2* KO mutants, cpDNA exhibited very similar link patterns (Fig. S11), indicating that most of these links are not artifacts.

Comment 27) lines 320-322: *Each of the recombination events (hr) in WT should be carefully examined as recombination across such short repeats are not expected in wild-type plants.*

Response) As described in the response to the comment 25, we think these recombination events show

basal level recombination in WT. We consider our ultra-deep sequencing can identify such low-level recombination products existing in WT.

Comment 28) Figure S11.

I don't understand the histograms? Are the result of windows analyzed? what is the 6M mean?

Response) Figure S11 is now Figure S12. We added detailed information to Fig. S12. They are window analysis, and 6M means 6 millions, and we replaced the note to 6×10^6 .

Comment 29) line 326: *"The number of recombination products without any homology at the junction (unk) was higher in the RECG KO mutant at hotspots for hr,"*

Do you mean there are not even short repeats? None longer than 3bp?

What to you meany by "hotspots for hr? please clarify.

Response) Yes, unk, the rearrangements without repeats longer than 3 bp was increased in the RECG KO cpDNA. However, these products are actually rearrangements occurred at very close regions, likewise deletion, as shown in Supp. Data 11. We haven't defined "hot sports for hr" before, and we rephrased this to "the clustering region between 30,000 and 60,000".

Revised) *The number of recombination products without any homology at the junction (unk) was higher in the RECG KO mutant at the clustering region between 30,000 and 60,000, but not in the RECA2 KO chloroplasts (Table 2 and Supp. Data 11).*

Comment 30) lines 346-347: *what do you mean by "hot spots"?*

The morisita index indicates if there is an even distribution or not. An uneven distribution does not imply the presence of hotspots.

Response) The citation was wrong, and was corrected.

We referred the region between 100,000-2,000 in which links are densely localized as a hot spot.

Revised) *p.17 RECG KO mtDNA links had a less biased distribution in terms of position (Table S10), outside of a few hot spots.*

E. coli analysis:

Comment 31) *The E.coli sequence used as reference is the same strains sequenced for wild-type and the mutants? why wild type shows rearrangements?*

A reference or plot of e. coli mapping pattern is missing.

Response) *E. coli* wild type genome sequence was used as a reference for mapping of genomes of all strains including mutants. Although the presence of repeats longer than 150 bps may disturb correct mapping of genomic rearrangements, our data show rearrangements of *E. coli* wild-type genome. We think

these rearrangements imply low-frequency rearrangements occurring in a population of the WT. However, we have omitted *E. coli* data as responded to the comment 1.

Comment 32) Figure 7A using a different scale than the one mentioned in the text (position $\sim 3.9 \times 10^6$ bp and a minimum at position $\sim 1.6 \times 10^6$ bp) and it is confusing.

Response) Thank you for the comment, we have omitted this figure in the revised manuscript.

Comment 33) lines 385-386: “whereas mutation in *recG* led to a sharp increase in read depth in the terminus area and a reduced population around *oriC* (Fig. 7A),”

I see the opposite.

Response) We meant sharp increase in read depth in the terminus area (1.5×10^6 bp) and a reduced population around *oriC* (3.9×10^6 bp) as compared to the WT. Please note that depth of *E. coli* genomes has not been normalized. We have omitted this figure in the revised manuscript.

Discussion:

Comment 34) lines 424: *I don't see evidence for asymmetric recombination products in the article on maize.*

Response) Asymmetric recombination in wild type should have cited Arabidopsis paper which is the same as citation for Arabidopsis mutants.

Revised) p.20) Accumulation of asymmetric recombination products is observed in *Arabidopsis* wild type and mutants as a hallmark of plant mitochondrial genome dynamics³².

Ref) 32. Davila JJ, *et al.* Double-strand break repair processes drive evolution of the mitochondrial genome in *Arabidopsis*. *BMC Biol* **9**, 64 (2011).

Comment 35) line 426: *what do you mean by crude analysis?*

Response) We meant rough analysis focusing on recombination between several specific repeats. We revised as follows.

Revised) p.20) This tendency was proposed based on a rough analysis of the *A. thaliana* *MSH1* mutant mitochondrial genome focusing on specific repeats which shows asymmetry of recombination at neighboring repeats³².

Comment 36) lines 442-444: “Second, the number of recombination products was not significantly correlated with the length of the repeats (Fig. S10),”

*Given that the maximum length of *P. patens* repeats is 79-90 bp and the fact that previously observed differences in recombination products are seen between large and intermediate-short repeats, what are the authors expecting from this analysis? Why would you expect any differences in repeats between 3 and*

90bp? It might be more interesting to compare repeats below 20bp (nonhomologous) and those above 25bp, which could be considered homologous.

Response) Our previous works showed higher accumulation level of products from recombination between 79-90 bp repeats that were detectable by DNA gel blot, while lower accumulation of products from recombination between short repeats (8-15 bp) that were detectable only by PCR (Odashima et al., 2009, 2015). We thus anticipated a correlation between length of repeats and accumulation level of the recombination products. We added description about these reasons. We have compared the number of recombination products between repeats below 20bp and above 21bp and found a difference between them. However, it is difficult to say the difference is significant because *P. patens* has small number of repeats longer than 20 bp, thus we avoided to discuss the comparison in the manuscript.

Revised) p.16) [Results] We anticipated a correlation between repeat length and read count of these mhr products, since previous studies on *P. patens* chloroplast showed higher and lower accumulation level of products from recombination between longer repeats (63 bp) and between short repeats (13-34 bp), respectively ²⁷.

Comment 37) Table S13; I would simply include the comparisons with the updated version of the cpDNA NC_005087.2 and in the revised version not even mentioned the "history" of first comparing with NC_005087.1 as it was the only available. Now, the version 2 is available and comparisons to the updated version should be included without mentioning the older version.

Response) We removed the three mutations from Table S11(renumbered) and replaced the NC_005087.1 to NC_005087.2

Finally, some of the responses were helpful to understand the reviewers' concerns but were not incorporated into the manuscripts. As many readers may have the same concerns, they should be included. Here are the two examples but please revise and also make sure you modify the text when responding to the reviewers.

Comment 38) The following statement provided to the reviewers should be incorporated in the text as no discussion of this topic is present in the current version:

"5. Also, I don't understand the .ins and .del files. In the mutant lines, did you observe <50bp indels? IF so, which mechanism would explain this indels?"

Response) ins and del are insertions and deletions shorter than 50 bp. Here ins include only the insertion by tandem repeat. We added the description of these in the revised text shown in the previous query. We also re-analyzed junction reads and revised Table 2. Since most of the in/dels are located in A/T cluster in cpDNA, these mutations should be derived from sequencing error or replication slippage that is a major source of spontaneous mutation in chloroplast (Massouh et al., Plant Cell, 2016). For mitochondria, we

observe the number of 'ins' is increased in *RECA1* KO-1 and the ins are mainly insertion of A, T, or AT, suggesting similar replication slippage in mitochondria. In the mutants, misannealing during recombinational repair may lead to increase of such indels. Because this result is unexpected and needs validation, we would like to avoid referring more about the indel in this paper. "

Response) We revised main text of the manuscript as follows.

Revised) p.16 [Results] **No mutant exhibited an apparent alteration in the number of insertions (ins) and deletions (del), which are mainly located in A/T cluster in cpDNA, or links indicating U-turn-like rearranged DNA molecules around palindromic sequence (pal) (Table 2 and Supp. Data 5-18).**

p.22 [Discussion] **In addition to recombination between SDRs, our analysis successively identified insertions and deletions in organelle DNA. Since most of the in/dels are located in A/T cluster in cpDNA, these mutations should be derived from replication slippage that is a major source of spontaneous mutation in chloroplast⁴¹. For mitochondria, we observe the number of 'ins' is increased in *RECA1* KO-1 and the ins are mainly insertion of A, T, or AT (Supp. Data 14), suggesting the occurrence of replication slippage similarly in mitochondria. Besides the higher copy number of cpDNA than mtDNA, such A/T cluster is more abundant in *P. patens* cpDNA than in mtDNA, resulting that the "del" was detected more in cpDNA. In the mutants, misannealing during recombinational repair may lead to increase of such indels.**

Comment 39) *The explanation for Fig.S1 could be included in the legend as it is likely that readers will have the same concerns.*

"6. Why in FigS1, the mapping rate of Illumina reads do not reach 100%?

Response) Thank you for the query. Fig. S1A shows the mapping rate of 150 bp Illumina reads allowing maximum of 2 bp mismatch per read, and they are about 86 % for all strains (numbers for each specific case can be found in Table S2). The mapping rate of reads truncated to 50 bp allowing up to 2 bp mismatch per read is shown in Fig. S1B and they are about 97% for all strains. These figures are to show the ratio of reads mapped to mt, cp and nc are similar with both mapping method, and to justify the estimate of mapping depth made with the mapping of truncated reads. The mapping rate of 150 bp reads with allowing maximum of 75 bp mismatch are about 99.6%, and we used this mapping method to carry out the junction read analysis. The paired read analysis was made using the truncated read mapping. And the mapping method with low mapping rate (h2r3) for 150 bp reads was not used for the rearrangement analysis. Part of the unmapped reads by the stringent method (h2r3) includes the junction reads, and the large portion of the rest of the unmapped reads should be the reads including sequencing errors."

Response) We added an explanation for the reason of the lower mapping rate of each strain in the caption of Fig. S1.

Revised) [Fig. S1 caption] **Supplementary Figure 1. Mapping of reads to nuclear, chloroplast, and mitochondrial DNA.**

- A. Mapping rate of Illumina reads (150 bp) with 2 bp mismatch permitted (h2r3).
B. Mapping rate of truncated reads (100 bp) with 2 bp mismatch permitted (chp100_h2r3).
Large portion of the unmapped reads in (A) and (B) should be reads including sequencing errors.

Comment 40) *The following should also be incorporated in the discussion:*

“15. Table 2, why so many “del” in cpDNA across WT and mutants? And why so few “del” in the mtDNA? Response) We found that most of these “del” are located at A/T cluster, thus this may be due to spontaneous deletion and/or sequencing error at A/T cluster. Besides the higher copy number of cpDNA than mtDNA, such A/T cluster is more abundant in P. patens cpDNA than in mtDNA, resulting that the “del” was detected more in cpDNA.”

Response) We have added the discussion in the main text as follows.

Revised) p.16) [Results] **No mutant exhibited an apparent alteration in the number of insertions (ins) and deletions (del), which are mainly located in A/T cluster in cpDNA, or links indicating U-turn-like rearranged DNA molecules around palindromic sequence (pal) (Table 2 and Supp. Data 5-18).**

p.22) [Discussion] **In addition to recombination between SDRs, our analysis successively identified insertions and deletions in organelle DNA. Since most of the in/dels are located in A/T cluster in cpDNA, these mutations should be derived from replication slippage that is a major source of spontaneous mutation in chloroplast⁴¹. For mitochondria, we observe the number of 'ins' is increased in RECAI KO-1 and the ins are mainly insertion of A, T, or AT (Supp. Data 14), suggesting the occurrence of replication slippage similarly in mitochondria. Besides the higher copy number of cpDNA than mtDNA, such A/T cluster is more abundant in P. patens cpDNA than in mtDNA, resulting that the “del” was detected more in cpDNA. In the mutants, misannealing during recombinational repair may lead to increase of such indels.**

Reviewer #2 (Remarks to the Author):

The authors have addressed most of my comments to this revised manuscript and I would like to reiterate that the results presented in this paper contribute to our better understanding of nature of organelle recombination. Together with the authors' thorough replies to the reviewer's comments, I am thinking that the present article is now ready for publication after minor revision as below.

Comment 1) *I do understand author's point of view on long-read sequencing from their reply. However, I would suggest that the authors should add their opinions on the long-read sequencing data in the*

manuscript, because the authors' opinions will help other readers why authors used short-read sequencing without using long-read sequencing. This will help readers to know that there is another approach (long-read based) to recombination analysis.

Response) We have added the explanation in the introduction section as follows.

Revised) p.23) [Discussion] In summary, we established a reliable and highly sensitive method to identify genome rearrangements by deep sequencing with short read. This revealed highly biased distribution of organelle genome rearrangements associated with production of various kinds of subgenomes. An alternative approach with long-read sequencing would give useful information to know both rearrangements and the detailed structure of such subgenomes, in the future study. Highly biased distribution of rearrangements suggests structural heterogeneity of organelle genomes, which should be related to the complex structure and replication of organelle DNA, both of which remain largely unknown. An integrative understanding of these mechanisms with HRR would solve the maintenance and dynamics of organelle genomes governed by HRR.

Comment 2) Typo in Figure 3

- 'map posision ~' → 'map position ~'
- '~ Paiered end ~' → '~ Paired end ~'
- '~ infered by' → '~ inferred by'

Response) Thank you for the comment. We have corrected them all.

Comment 3) Figure legends in the main text and the figures do not match. Please correct it.

- (e.g.) Figure 1 in the main text: "Detection of rearrangement by (a) paired-end reads and (b) junction reads"; Figure 1 in the figures: "Ratio of mapped reads"

Response) Thank you for the comment. We have checked all the figure legends and revised them.

Comment 4) [Page 4, Line 73] The authors have stated "A. thaliana has three RECAs, which are localized to the chloroplast (AtRECA1), chloroplast and mitochondria (AtRECA2), and mitochondria (AtRECA3). Note that gene name and localization of their products do not correspond between P. patens and A. thaliana regarding RECA genes." in revised manuscript based on Reviewer 1's comment. However, determining correspondence between P. patens and A. thaliana RECA genes using only locality information isn't reasonable in my opinion. I would like to suggest a simple phylogenetic analysis to see a relationship between P. patens and A. thaliana based on RECA genes.

Response) Phylogenetic tree of plant RECA genes including those of P. patens and A. thaliana has already been tested by our group, and the tree clearly show the relationship between RECA genes of P. patens and A. thaliana. We cited this and revised main text correspondingly.

Revised) p.4) *A. thaliana* has three RECA, which are localized to the chloroplast (AtRECA1), chloroplast and mitochondria (AtRECA2), and mitochondria (AtRECA3)¹⁰. **Note that gene names and localization of their products as well as their phylogenetic relationship do not correspond between *P. patens* and *A. thaliana* regarding RECA genes^{13, 24}.**

Comment 5) Reference format is not consistent. Please follow the format of *Communications Biology*.

Response) We followed the reference format of *Communication Biology* for all the references.

Comment 6) I don't think it's necessary to place Figure 2 ("Ratio of mapped reads") in the main figure.

Response) Thank you for the comment. Ratio of mapped reads shown as Figure 1 of current version of our manuscript shows an important phenotype of the mutants that is difficult to understand from other mapping data such as depth. Therefore, we would like to show this figure as a main figure of our manuscript.

Comment 7) Supplementary Table 11: I would like to suggest changing Q20 (99% accuracy) to Q30 (99.9% accuracy) in this table as Supplementary Table 1.

Response) Thank you for the comment. In response to the comments from reviewer 1, we have omitted *E. coli* data from the manuscript.

REVIEWERS' COMMENTS:

Reviewer #1 (Remarks to the Author):

The authors have addressed all the comments and suggestions presented by the reviewers and the manuscript is much improved.

In particular, I'm pleased to see that the analyses of truncated reads have been re-done by changing the truncated reads from 50 to 100bp.

Minor comments:

lines 120-122: "reveal" is used twice in this sentence.

line 219: correct grammar: "if there found" is incorrect.

line 223: "thereby number" should be changed to "thereby the number "

lines 226-230: "Homologous rearrangements within close vicinity (< 70 bps) in opposite direction are categorized as (5) U-turn like rearrangements (pal; Fig. S7). The homologous sequence for these rearrangements can be observed as a palindrome, or quasi palindrome sequence. Data S8~S18 lists all rearrangements (ins, d 229 el, mhr, unk, pal, mhmr) detected in our analysis."

"Homologous rearrangements" should be changed to "Rearrangements " alone, because there is no evidence that all these rearrangements involved a homologous sequence.

"The homologous sequence for these rearrangements can be observed as a palindrome,": same here, there is no evidence that these palindromic sequences are homologous. Delete "homologous" in the sentence.

lines 237-239: improve grammar.

line 242: to support for the, delete "for".

lines 328: Rephrase: "... in longer (63 bp) and shorter (13-34 bp) repeats, respectively."

lines 335: "The number of recombination products without any homology at the junction (unk)" should be changed to "The number of recombination products without any SEQUENCE SIMILARITY at the junction (unk)".

Reviewer #2 (Remarks to the Author):

The authors properly addressed most of earlier comments, which I suggested. I agree with Reviewer 1's comments about omitting E. coli data from the manuscript. Together with the authors' thoroughness in addressing the first reviewer's insightful comments, I believe the present manuscript will now be ready for publication.

Minor comments:

1. In supplementary methods, the authors stated that "... we downloaded genome sequence of chloroplast (NC_005087.2: 122,890bp) and mitochondrion (NC_007945.1: 105,340bp) from NCBI.'. But in main text (p26, lines 535-537), the authors stated that "... we used harbored several mutations in its organelle DNA relative to registered chloroplast [NC_005087.1] and mitochondrial genomes [NC_007945.1] (Table S11).' Please check this detail (chloroplast genome) and update it to the correct version.

Response to reviewers' comments

We would like to thank the reviewers for their valuable comments. In response to the reviewers' comments, we have revised our manuscript. We believe that the revised version of the manuscript is now sufficient for publication in *Communications Biology*.

REVIEWERS' COMMENTS:

Reviewer #1 (Remarks to the Author):

The authors have addressed all the comments and suggestions presented by the reviewers and the manuscript is much improved.

In particular, I'm pleased to see that the analyses of truncated reads have been re-done by changing the truncated reads from 50 to 100bp.

Minor comments:

Comment 1) lines 120-122: "reveal" is used twice in this sentence.

Response) We have revised the sentence as follows.

The deep sequencing data revealed the distribution in read depth of organelle DNA, which are implicated as the consequences of genome instability and genome rearrangement.

Comment 2) line 219: correct grammar: "if there found" is incorrect.

Response) We have revised the sentence as follows.

Rearrangement between two distant locations on reference are categorized as (3) Rearrangement based on microhomology/homology (mhr) if **there are** any identical sequences longer than 3 bps responsible for the rearrangement, and (4) Rearrangement without any **sequence similarity** (unk), if **there is** no homologous sequence around the junction points.

Comment 3) line 223: "thereby number" should be changed to "thereby the number "

Response) Revised.

Comment 4) lines 226-230: "Homologous rearrangements within close vicinity (< 70 bps) in opposite direction are categorized as (5) U-turn like rearrangements (pal; Fig. S7). The homologous sequence for these rearrangements can be observed as a palindrome, or quasi palindrome sequence. Data S8~S18 lists all rearrangements (ins, d 229 el, mhr, unk, pal, mhmr) detected in our analysis."

"Homologous rearrangements" should be changed to "Rearrangements " alone, because there is no evidence that all these rearrangements involved a homologous sequence.

*"The homologous sequence for these rearrangements can be observed as a palindrome,":
same here, there is no evidence that these palindromic sequences are homologous. Delete "homologous" in the sentence.*

Response) Revised as follows.

Rearrangements within close vicinity (< 70 bps) in opposite direction are categorized as (5) U-turn like rearrangements (pal; Supplementary Fig. 7). **The sequence** for these rearrangements can be observed as a palindrome, or quasi palindrome sequence. Data S8~S18 lists all rearrangements (ins, del, mhr, unk, pal, mhmr) detected in our analysis.

Comment 5) lines 237-239: improve grammar.

Response) Revised as follows.

Nevertheless, the high matching ratio **suggests** rearrangements detected by our junction read method are reliable.

Comment 6) line 242: to support for the, delete "for".

Response) We have deleted the "for".

Comment 7) lines 328: Rephrase: "... in longer (63 bp) and shorter (13-34 bp) repeats, respectively."

Response) Revised as follows.

We anticipated a correlation between repeat length and read count of these mhr products, since previous studies on *P. patens* chloroplast showed higher and lower accumulation level of products from recombination **between 63-bp repeats and between 13-34 bp repeats**, respectively²⁷.

Comment 8) lines 335: "The number of recombination products without any homology at the

junction (unk)" should be changed to "The number of recombination products without any SEQUENCE SIMILARITY at the junction (unk)".

Response) Revised.

Reviewer #2 (Remarks to the Author):

The authors properly addressed most of earlier comments, which I suggested. I agree with Reviewer 1's comments about omitting E. coli data from the manuscript. Together with the authors' thoroughness in addressing the first reviewer's insightful comments, I believe the present manuscript will now be ready for publication.

Minor comments:

1. In supplementary methods, the authors stated that '... we downloaded genome sequence of chloroplast (NC_005087.2: 122,890bp) and mitochondrion (NC_007945.1: 105,340bp) from NCBI.'. But in main text (p26, lines 535-537), the authors stated that '... we used harbored several mutations in its organelle DNA relative to registered chloroplast [NC_005087.1] and mitochondrial genomes [NC_007945.1] (Table S11).' Please check this detail (chloroplast

genome) and update it to the correct version.

Response) We have updated accessions of organelle DNA sequences used in both main and supplementary methods and they are now integrated into the Methods of main text.